# Physiological and Proteome Analysis of the Effects of Chitosan Oligosaccharides on Salt Tolerance of Rice Seedlings

**DOI:** 10.3390/ijms25115953

**Published:** 2024-05-29

**Authors:** Xiangyu Qian, Yaqing He, Lu Zhang, Xianzhen Li, Wenzhu Tang

**Affiliations:** School of Biological Engineering, Dalian Polytechnic University, Dalian 116034, China; 17604269303@163.com (X.Q.); hyq34448464@163.com (Y.H.); zhanglu967999@163.com (L.Z.); xianzhen@dlpu.edu.cn (X.L.)

**Keywords:** Nipponbare, rice seedlings, chitosan oligosaccharide, salt stress, proteomics

## Abstract

Rice (*Oryza sativa* L.) is an important social-economic crop, and rice seedlings are easily affected by salt stress. Chitosan oligosaccharide (COS) plays a positive role in promoting plant growth and development. To gain a better understanding of the salt tolerance mechanism of rice under the action of COS, Nipponbare rice seedlings were selected as the experimental materials, and the physiological and biochemical indexes of rice seedlings in three stages (normal growth, salt stress and recovery) were measured. Unlabelled quantitative proteomics technology was used to study differential protein and signaling pathways of rice seedlings under salt stress, and the mechanism of COS to improve rice tolerance to salt stress was elucidated. Results showed that after treatment with COS, the chlorophyll content of rice seedlings was 1.26 times higher than that of the blank group (CK). The root activity during the recovery stage was 1.46 times that of the CK group. The soluble sugar in root, stem and leaf increased by 53.42%, 77.10% and 9.37%, respectively. The total amino acid content increased by 77% during the stem recovery stage. Furthermore, the malondialdehyde content in root, stem and leaf increased by 21.28%, 26.67% and 32.69%, respectively. The activity of oxide dismutase (SOD), peroxidase (POD) and oxygenase (CAT) were increased. There were more differentially expressed proteins in the three parts of the experimental group than in the CK group. Gene Ontology (GO) annotation of these differentially expressed proteins revealed that the experimental group was enriched for more entries. Then, through the Kyoto Encyclopedia of Genes and Genomes (KEGG), the top ten pathways enriched with differentially expressed proteins in the two groups (COS and CK groups) were utilized, and a detailed interpretation of the glycolysis and photosynthesis pathways was provided. Five key proteins, including phosphofructokinase, fructose bisphosphate aldolases, glycer-aldehyde-3-phosphate dehydrogenase, enolase and pyruvate kinase, were identified in the glycolysis pathway. In the photosynthesis pathway, oxygen evolution enhancement proteins, iron redox proteins and ferredoxin-NADPH reductase were the key proteins. The addition of COS led to an increase in the abundance of proteins, a response of rice seedlings to salt stress. COS helped rice seedlings resist salt stress. Furthermore, using COS as biopesticides and biofertilizers can effectively increase the utilization of saline-affected farmland, thereby contributing to the alleviating of the global food crisis.

## 1. Introduction

Rice is one of the most important food crops in the world. Rice cultivated in Asia was domesticated from its wild ancestor (*Oryza rufipogon* Griff). About one-half of the world’s population feed on rice. Rapid global population growth and the damage caused by the climate crisis have led to a significant decline in rice production [1]. Salinity and drought are the two main stressors that directly affect crop yield. Approximately 20% of the world’s irrigated agricultural lands are adversely affected by soil salinization [2]. Salt stress is an abiotic stress that affects normal growth and production, and high salinity can lead to a reduction of 30–80% in rice production [3]. Salt stress causes many aspects of rice defense reactions, such as changes in cell structure, physiology, shape and structure, biochemical synthesizing and activation of other molecules [4,5,6]. Rice is often severely affected by salt stress worldwide. The most sensitive period to salt stress is during the seedling stage, and the subsequent vegetative growth period will improve its tolerance to salt stress [7,8]. Therefore, to improve rice yield, increasing the salt tolerance of rice seedlings is a key factor.

Chitosan is a linear polysaccharide derived from the N-deacetylation of chitin, containing varying degrees of N-acetylation. It consists primarily of β-(1,4)-linked 2-acetamido-D-glucopyranose and 2-amino-D-glucopyranose units, with the latter typically comprising more than 80% of the structure [9]. Chitosan is naturally synthesized by fungi in their cell walls. COS is a type of chitosan with a low degree of polymerization (≤20) and an average molecular weight below 3900 Da, usually falling within the range of 0.2–3.0 kDa [10]. COS has attracted wide attention because of its low molecular, short chain structure, diverse bioactivities and wide application in different fields [11,12]. COS can induce plants to activate defense response under biotic or abiotic stress, including the ability to effectively inhibit fungal growth, regulate and improve the activity of the defense enzyme system, and promote the production of favorable metabolites in plants, so it can be used as a plant immune agent in agricultural production [13].

As a modern science technology to study all proteins and explore the chemical composition of genome-encoded proteins, proteomics can be used to discover the causes of different physiological characteristics in organisms with the help of multiple technologies, such as two-dimensional electrophoresis, mass spectrometry, chromatography, etc. [14,15]. For example, Sengupta utilized proteomics to uncover over 400 protein spots that significantly changed under salt stress across different salt-tolerant rice germplasm. Among these differentially expressed proteins, a highly up-regulated protein in wild salinity-tolerant rice, named SSP8908, was identified as a cellulose synthase-like protein under 400 mM NaCl stress. SSP8908 likely serves as a crucial factor in regulating cellulose synthesis in the cell walls of wild rice, thereby potentially enhancing its salt tolerance [16]. Liu conducted a proteomic analysis to compare ubiquitin-related proteins involved in salt sensitivity in the root systems of the TNG67 rice variety and the NaN_3_-induced pure mutants SM75 (salt-tolerant) and SA0604 (salt-sensitive). The results indicated that under salt stress, ubiquitination of cellulose synthase α increased in TNG67 and SM75 but decreased in SA0604. Moreover, the salt tolerance of these lines correlated with the abundance of ubiquitinated cellulose synthase α under salt stress, further highlighting the significant role of cellulose in the salt tolerance of rice [17]. Consequently, proteomic studies are helpful in revealing changes in protein abundance and the effects of different protein modifications in plants under various stresses.

*Oryza Sativa* L. spp. japonica, var Nipponbare serves as a crucial model organism for the study of rice genetic material. Through sequential cloning techniques, Nipponbare is the only rice variety among monocots whose genome has been sequenced with high quality. The completion of its genome sequencing has facilitated further study of rice gene functions, contributing to food security [18]. Therefore, this experiment was performed using rice (Nipponbare) as an experimental material, and a comparison was made between the experimental group (COS) sprayed with COS and the CK group without COS. The effects of salt stress on rice were studied by measuring the physiological and biochemical indexes under three physiological conditions in the COS group and the blank group. The measured physiological and biochemical indicators included chlorophyll, soluble sugars, malondialdehyde, root iability, total amino acid content, SOD, POD and CAT activity. The label-free proteomics method was used to investigate the abundance of proteins in roots, stems and leaves in rice seedlings under salt stress. Enrichment analysis was then performed using differential protein expression, and protein functions and pathways were analyzed using Gene Ontology and the Kyoto Encyclopedia of Genes and Genomes. Finally, the above analysis of protein changes proved that the addition of COS enabled rice itself to resist the adverse effects of salt stress.

## 2. Results and Discussion

### 2.1. Effects of COS on Rice Seedlings under Salt Stress

#### 2.1.1. Growth and Morphological Characteristics of Rice Seedlings

According to the phenotypic observation, the rice seedings in normal growth were pale green, and the stems and leaves of the seedlings were slender (Figure 1A,D). Under 150 mM NaCl stress, yellowing and lodging began to appear in seedings (Figure 1B,E). In comparison to the COS group, the CK group showed more pronounced lodging and yellowing. In the recovery stage, as shown in Figure 1C,F, rice tips and stems in both groups were more yellow than during the salt stress stage. Meanwhile, the lodging and leaf curl degree in the CK group were more serious and the stems were thinner. The COS group grew better and the thick stems could support the long leaves without serious lodging phenomenon. Therefore, it could be seen that spraying COS could improve the harm of salt stress to rice on some extent.

During the recovery period of rice seedings, contrasted with the CK group, the roots length, leaf length and leaf width of the COS group increased by 16.77%, 10.43% and 20.83%, respectively. In comparison to the CK group, the group treated with COS exhibited increases in fresh weight for roots, stems and leaves by 7.41%, 14.04% and 3.45%, respectively. Compared with the CK group, in the COS group, the dry weight of the roots, stems and leaves increased by 5%, 31.37% and 1.27%, respectively (Table 1). These results are consistent with previous studies showing that COS can promote the budding growth of rice seedlings [19,20]. In the three processes of rice seedling growth, COS spraying could increase the fresh weight of all three parts of rice. In addition, there were significant differences in the normal growth and recovery stage. It indicated that the fresh weight of rice seedlings would be damaged to a certain extent after salt stress, but the addition of COS alleviated the loss caused by salt stress. The data in Table 1 also showed that the dry weight of roots, stems and leaves of rice with COS added was higher than that of CK group. During the whole growth process, the dry weight of the stem in the CK group was lower than that of the COS group, but there was no significant difference between the roots and leaves. In the salt stress and recovery stage, there was a very significant difference between the stems in the COS group and the CK group, indicating that the addition of COS alleviated the damage of rice seedlings caused by stress. The results align with Liu and Mostek’s research, showing that salt stress could affect the dry and fresh weights of roots, stems, and leaves, and that after salt stress, both the dry and fresh weights of rice seedlings increased [21,22].

#### 2.1.2. Biochemical Indicators of Rice Seedlings

To elucidate possible mechanisms leading to salt stress tolerance after treatment with COS, the changes in physiological indexes related to salt stress were analyzed, including the content of chlorophyll, soluble sugar, malondialdehyde (MDA), root activity, total amino acid and the activity of three main antioxidant enzymes (SOD, POD, CAT).

In both the COS group and the CK group, the trend of chlorophyll content increased first and then decreased. The small amount of increase in the salt stress stage may be related to the continued growth of rice seedlings, although they were damaged by salt stress. However, during the recovery stage, it could be seen that chlorophyll decreased in both groups because of the obvious yellowing of rice leaf tips, which was consistent with the above observations on the phenotype of rice seedlings. The chlorophyll content of all COS groups was higher than that of the CK group, indicating that spraying COS improved the photosynthesis of rice leaves (Figure 2A). After adding COS, the root activity was higher than that of untreated rice, and there were significant differences in salt stress and recovery stage. It indicated that the addition of COS could enhance the root activity of rice seedlings under salt stress to maintain their normal growth (Figure 2B). The content of soluble sugar content in the roots of the CK group gradually increased with the growth time, while that in the stems and leaves decreased first and then increased. However, during the salt stress stage of roots, the soluble sugar content of the COS group was nearly twice more than that of the CK group, indicating that the addition of COS in rice could provide more nutrients and energy to resist the stress (Figure 2C–E). The MDA content of the CK group was higher than that of the COS group. The MDA content in both groups increased in roots and leaves with stress time. The MDA content of the stems decreased a bit more in the recovery stage than in the normal stage, possibly related to plant site regulation (Figure 2F–H). The total amino acid content of the three parts changed differently, but that of the COS group was higher than that of the CK group, and there were significant differences, indicating that the addition of COS could enhance the salt resistance of rice (Figure 2I–K). The SOD, POD and CAT activities of the COS groups of root, stem and leaf were all greater than those of the CK group, and the differences were significant (Figure 2L–T). The findings align with those of Bai et al. [23], who suggested that under salt stress conditions, oat roots displayed increased levels of SOD, POD, MDA and soluble sugar content. SOD can catalyze superoxide radicals to hydrogen peroxide (H_2_O_2_) and POD can scavenge H_2_O_2_. CAT can protect cells from H_2_O_2_ damage by catalyzing the decomposition of H_2_O_2_ into O_2_ and H_2_O. In general, the enhanced activities of SOD, POD and CAT were rice’s defense responses to long-term salt stress, and the addition of COS enhanced these defense responses.

### 2.2. Label-Free Quantitative Proteomics Analysis

The samples of roots, stems and leaves of rice seedlings were tested in the three stages of normal growth, salt stress and recovery, respectively. There were 18 different groups of samples, and each group of samples was tested three times. The following table shows the number of proteins identified by mass spectrometry in the rice seedlings of Nipponbare (Table 2). Besides the salt stress stage of root, the number of proteins identified was higher in the COS group than in the CK group. This indicated that the addition of COS was able to increase protein expression in rice seedlings.

The stably expressed proteins with CV < 20% and the differentially expressed proteins (DEPs) were screened by Label-free Quantitative (Table 3). In the three stages of leaves in COS group, there were 2427 stable proteins and 1476 DEPs (1006 up-and 470 down-regulated) in S/N. In the R/N, there were 2236 stable proteins and 1650 DEPs (1060 up- and 590 down-regulated). In the R/S, 1460 proteins were found to be stable, while 1687 proteins showed differential expression (1027 up-and 660 down-regulated). In the three stages of stems in COS group, there were 3227 stable proteins and 1857 DEPs (1082 up-and 775 down-regulated) in S/N. A total of 2853 stable proteins and 1897 DEPs (922 up-and 975 down-regulated) were identified in R/N. In the R/S, 2976 proteins were found to be stable, while 1806 proteins showed differential expression (829 up-and 977 down-regulated). In the three stages of roots in COS group, there were 1595 stable proteins and 1252 DEPs (435 up- and 817 down-regulated) in S/N. In the R/N, there were 1084 stable proteins and 855 DEPs (258 up-and 597 down-regulated). A total of 1316 stable proteins and 778 DEPs (335 up-and 443 down-regulated) were identified in R/N.

In the three parts of the COS group, the number of up-regulated proteins in the three stages of leaves was approximately double that of the down-regulated proteins. There was no significant difference between the number of up-regulated and down-regulated proteins in stems at three stages. However, in the roots, the number of down-regulated proteins was significantly higher than that of up-regulated proteins across the three stages. This showed that under salt stress, the protein expression in rice seedlings shows positional differences, and salt stress also caused significant changes in rice protein expression in the whole plant.

A Venn diagram of differential proteins in the three parts of the rice seedlings is presented in Figure 3. In the COS group, a total of 673 DEPs were identified in the three stages of leaves. Of these, in S/N (salt stress/normal growth) and R/S (recovery/salt stress), a total of 305 DEPs were identified, and in R/N and R/S, a total of 305 DEPs were also identified, whereas in R/N and R/S, a total of 167 DEPs were identified. The total number of the three stages of DEPs in the CK group was about the same as that of the COS group. But only 83 DEPs were shared by S/N and R/N, much less than the COS group. In the stems, 637 DEPs were shared under the three stages. The shared DEP counts were 457 for S/N and R/N, 537 for R/N and R/S, and 298 for S/N and R/S. 457 DEPs were shared between S/N and R/N, 537 between R/N and R/S, and 298 between S/N and R/S. Each crossover section had about twice as many as the CK group. In the roots, there were 349 DEPs shared under the three stages, 283 DEPs shared by S/N and R/N, 102 DEPs shared by R/N and R/S, and 74 DEPs shared in S/N and R/S. The average DEPs in the CK group was nearly half less than that in the COS group, and the number of S/N and R/N groups was quite different. Therefore, by observing the common DEPs across each stage, it could be supposed that after the application of COS, various parts of the rice plant could resist the damage caused by salt stress through the expression of common proteins.

### 2.3. GO Annotation Analysis

To further determine the function of the DEPs, a GO annotation analysis was performed, and biological process (BP), cell component (CC) and molecular function (MF) were included.

The top 15 items in the BP were anatomical structure development, biosynthesis process, catabolic process, cell component organization or biosynthesis, cell metabolic process, cell response to stimuli, establishment of localization, nitrogen compound metabolism, organic metabolism, primary metabolic process, biological process regulation, response to abiotic stimuli, response to chemical substances, response to stress and small molecule metabolic process. The top 15 items in the CC were catalytic complex, cell junction, cell periphery, cytoplasm, cytosol, inner membrane system, nuclear membrane, inner cellular anatomy, membrane, membrane lumen, organelles, organelle compartments, plastid matrix, ribonucleoprotein complex and symplast. The top 15 items in MF were carbohydrate derivative binding, catalytic activity, protein, catalytic activity, RNA, drug binding, heterocycle compound binding, hydrolase activity, ion binding, ligase activity, organic cyclic compound binding, oxidoreductase activity, protein binding, small molecule binding, structural components of ribosome, transferase activity and transmembrane transporter activity.

The GO annotation of differential proteins in rice leaves is shown in Figure 4. In the S/N, the addition of COS resulted in more annotations than the CK group in 14 terms in BP, 12 terms in CC and 15 terms in MF, respectively. In the R/N, the addition of COS resulted in more annotations than the CK group in 8 terms in BP, 14 terms in CC and 12 terms in MF, respectively. In the R/S, the addition of COS resulted in more annotations than the CK group in 10 terms in BP, 7 terms in CC and 7 terms in MF, respectively.

In the stems, S/N had fewer catabolic processes and more endogenous stimulus responses in the BP top 15 entries. In CC, the first 15 entries did not differ in the three stages. The CK group did not differ from the COS group in the first 15 CC entries. In stems, the addition of COS in the S/N resulted in more annotations than the CK group in 9 terms in BP, 12 in CC and 11 in MF (Figure 5). In the R/N, all annotations in the COS group were more than those in CK group. After addition of COS, in the R/S, there were more annotations in 8, 10 and 5 terms in BP, CC and MF, respectively.

In the roots, the number of enrichments in the top 15 GO annotations of BP, CC and MF corresponding to the S/N, R/N and R/S in the COS group were all more than those in CK group (Figure 6). In the BP top 15 entries, there was less established localization and more endogenous stimulus responses in the R/N. In the CC top 15 entries, there was less collective matrix in R/S. In the MF top 15 entries, R/S and the other two stages had less ligase activity and lyase activity, and more enzyme regulation activity and isomerase activity. Among them, S/N has less catalytic activity, acts on RNA and drug binding, and has more enzyme regulation activity and isomerase activity.

Overall, the addition of COS to rice seedlings could increase the number of differential proteins, indicating that COS may act as modulators for rice encountering salt stress. Especially for the stems and roots, the number of DEPs by GO annotations enriched in the COS group was significantly more than that in the CK group.

From the GO annotation map of the above three sites, it can be seen that the top 15 annotations were not very different, and the differences were also the result of the regulation of plants in the site and stage. It showed that after rice seedlings were subjected to salt stress, in order to resist the damage from salt stress, its vital activities have changed, which in turn led to alterations in annotations.

### 2.4. KEGG Metabolic Pathway Analysis

To gain an insight into the main biological functions and associated metabolic processes of the DEPs of the COS group, a KEGG analysis was performed. The top 10 critical pathways that were enriched in DEPs of rice seedlings are depicted in Figure 7. In the S/N, R/N and R/S of the leaves of the COS groups, there were 7, 9 and 7 signaling pathways, respectively, with a more enrichment of DEPs compared to the CK group. In the S/N, R/N and R/S of the stems of the COS groups, there were 7, 10 and 10 signaling pathways, respectively, with a more enrichment of DEPs compared to the CK group. In the S/N, R/N and R/S of the roots of the COS groups, there were 10, 10 and 9 signaling pathways, respectively, with a more enrichment of DEPs compared to the CK group. It was found that after the addition of COS, the DEPs enriched in the top 10 key pathways of rice seedlings were significantly increased. Therefore, COS has positive effects on the physiological response and metabolic pathways of rice seedlings. The following section provides a detailed discussion of DEPs produced in the glycolysis and photosynthesis pathways.

#### 2.4.1. Glycolysis Pathway

Glycolysis can directly participate in plants’ response to environmental stresses, such as salt stress, drought and cold stress. As shown in Figure 7, the number of DEPs identified in the glycolysis pathway of rice seedlings with COS was higher than that of the CK group, which indicated that COS promoted expression of more proteins to resist salt stress. Moreover, DEPs had different levels of response in the normal growth stage, salt stress and recovery stage. In the leaves of rice seedlings, there were 12 co-expressed proteins in the COS group and CK group. The abundance of 6, 11 and 2 proteins in the COS group was higher than that in the CK group at the normal growth, salt stress and recovery stage, respectively. In the stems, 6 proteins were co-expressed in the two groups. The abundance of 4, 4 and 3 proteins in the COS group was higher than that in the CK group at the normal growth, salt stress and recovery stage, respectively. There were 8 co-expressed proteins in the roots of the two groups. The abundance of 7, 2 and 3 proteins in the COS group was higher than that in the CK group at the normal growth, salt stress and recovery stage, respectively (Table 4). According to the data, the expression of the common proteins was up-regulated by spraying COS under normal growth. The results showed that the addition of COS could enhance the regulation of glycolysis.

Among the main glycolysis pathways in this experiment, phosphofructokinase (EC2.7.1.90), fructose bisphosphate aldolases (EC4.1.2.13), glyceraldehyde-3-phosphate dehydrogenase (EC1.2.1.12), enolase (EC4.2.1.11) and pyruvate kinase (EC2.7.1.40) were key proteins. There were many reports about the glycolysis pathway of plants under stress. For example, Liu et al. [19] reported that proteins involved in glycolysis are inhibited in the presence of certain stressors in proteomic analysis of early salt stress responses in rice roots and leaves. However, after salt stress, the expression of various enzymes in rice will be adjusted to maintain the growth of rice. Therefore, in the present study, phosphofructokinase in roots was up-regulated after salt stress and it could serve as a marker for early signals of salt stress. In the results of proteomic analysis in response to early salt stress and the changes in plant proteome caused by salt stress, fructose bisphosphate aldolases were up-regulated after encountering salt stress [24,25]. Increased accumulation of fructose bisphosphate aldolases was also frequently observed in the rice sheath after salt stress [26]. In the present study, fructose bisphosphate aldolases were found as common proteins in leaves, stems and roots, with the same trend of up-regulation. Du et al. [27] found that glyceraldehyde-3-phosphate dehydrogenase was up-regulated in cucumbers under salt stress. In the present study, the protein abundance of glyceraldehyde-3-phosphate dehydrogenase in leaves increased significantly in both the CK group and COS group, which was consistent with the report. The study by Du et al. showed that under salt stress, the expression of enolase in cucumber roots increased, thereby enhancing its glycolytic activity [27]. It was the same result in this present study that both groups of enolases were expressed and up-regulated after salt stress. Pyruvate kinase is an important regulatory enzyme in the glycolytic metabolism, catalyzing the final step of glycolysis. It irreversibly converts phosphoenolpyruvate into pyruvate and is one of the key enzymes in the glycolytic pathway. Studies on *Arabidopsis thaliana* have shown that under unstressed and salt stressed conditions, protein signaling is affected. The expression of pyruvate kinase in *A. thaliana rosettes* is up-regulated when chitosan oligosaccharides are added under salt stress [28]. In the COS group of this experiment, the protein abundance of pyruvate kinase increased from normal growth to salt stress. This is because the glycolysis was enhanced by spraying COS, thus enabling plants to resist the damage caused by salt stress.

The DEPs in the COS group are shown in Table 5. There were 14 DEPs in the S/N and R/N of leaves (8 DEPs in S/N and 6 DEPs in R/N). In the stem, there were no DEPs in S/N and 11 in R/N. There were 23 DEPs in the S/N and R/N of roots (12 DEPs in S/N and 11 DEPs in R/N). The activation of glycolysis repairs the damage caused by salt stress in rice seedlings by providing an increased expression of proteins. Additionally, the expression of DEPs in rice seedlings varied in different parts, indicating that the regulatory mechanism of rice was not the same. Overall, the changes in rice seedlings under salt stress in the glycolysis pathway mainly involve the expression of common proteins.

#### 2.4.2. Photosynthesis Pathway

In plants, the light reactions occur in the chloroplasts, where light energy is utilized by photosystem I and II. The NADPH produced after the reactions is used by ATP synthase to generate ATP. Subsequently, the supply of ATP and NAD(P)H is used to fix carbon dioxide. Plant photosynthesis can produce carbon sources (such as glyceraldehyde-3-phosphate, etc.). These carbon sources are further transformed into energy-rich substances such as glucose, sucrose and starch through light and dark reactions in chloroplasts. In most plants, excessive salt mediated by roots closes stomata in leaves and hinders carbon dioxide assimilation. Therefore, the level of salt will affect plant photosynthesis and then affect the accumulation of organic matter in the process of plant growth. Then, we analyzed the differential proteins identified in the photosynthetic pathway of rice seedlings after adding COS.

There were 13 co-expressed proteins in leaves in the COS group and CK group, as shown in Table 6. The abundance of 9, 9 and 5 proteins in the COS group was higher than that in the CK group at the normal growth, salt stress and recovery stage, respectively. In the stem, there were 7 co-expressed proteins in the two groups. The abundance of 4, 1 and 5 proteins in the COS group was higher than that in the CK group at the normal growth, salt stress and recovery stage, respectively. In leaves, most of the photosynthesis proteins in the COS group were more abundant than those in the CK group. In the stems, during the salt stress phase, only one common protein was up-regulated in the COS group. However, during the recovery stage, the number of proteins with more abundance increased. This suggests that under salt stress, the regulatory mechanism of photosynthetic proteins in the stems may differ from that in the leaves, because photosynthesis primarily occurs in the leaves. From the data of the two groups, even if the salt affected the photosynthesis of plants, the addition of COS could regulate the abundance of common proteins, thus enhancing photosynthesis.

Photosystem II supercomplex is a protein–pigment system that exists in the thylakoid membranes of advanced plants. This system is divided into two distinct subcomplexes. The first is the light-harvesting complex (LHC), which is involved in light capture and energy distribution. The second is the PSII core, which includes the PSII reaction center (RC) and the oxygen-evolving complex (OEC) and is associated with water splitting and oxygen generation. The data of this experiment showed that among the common proteins in leaves of the COS group, oxygen-evolution-enhancing proteins PsbO, PsbP and PsbQ were detected, and most of them were up-regulated under salt stress. This was consistent with the results identified by Kim et al., where the oxygen-evolving proteins (OEPs) of PSII and protein 2 of the PSII oxygen-evolving complex significantly increased under salt stress [29]. In addition, Martino et al. also proposed the view that salt stress may enhance photorespiration [30]. Under salt stress, photosystem II needs to increase its expression level to oxidize water, release oxygen and transfer electrons to counteract the adverse effects of the stress. Among the common proteins in the stem of COS group, the abundance of PsbS light-harvesting chlorophyll a/b binding protein was increased. This indicated that maintaining the level of photoreactive protein allowed sufficient energy transfer under salt stress, which was the same as the results of Frukh et al. [31].

Changes in protein abundance in photosystem I may regulate the efficiency of electron transfer and the transmembrane electrochemical proton gradient, thereby affecting ATP synthesis and the formation of NADPH. In salt-stressed plants, multiple isoforms of the chloroplast ATP synthase and the iron protein NADPH oxidoreductase (FNR) are regulated by salinity. In this experiment, the common proteins in leaves and stems of rice seedlings in the COS group included PetF and PetH of photosystem I, namely ferredoxin and ferredoxin-NADPH reductase. The abundance of PetH protein in leaves was up-regulated by salt stress, which ensured adequate energy for other processes important during salt stress. The abundance of PetF was down-regulated, and ferredoxin was a very strong reductant, which may be due to the damage to the photosystem I complex by salt stress. Studies on the photosynthetic reaction-related proteins (PetF and PetH) in *Arabidopsis thaliana* under salt stress indicate that the expression of photoreaction proteins in plants is regulated under salinity conditions to maintain the normal growth rate of plants [32].

In the COS group, there were 14 DEPs in the S/N and R/N (8 DEPs in S/N and 6 in R/N) in leaves. In the stems, there were 24 DEPs (6 DEPs in S/N and 18 in R/N) (Table 7). Among them, the DEPs in the leaves of the COS group were mainly concentrated in the ATP synthase and optoelectronic transport sites. The DEPs in the stems of the COS group were mainly concentrated in photosystem II under S/N, and the photosystem II and ATP synthase under R/N. This indicated that the regulation of photosynthesis in the stems of the COS group was mainly different in the photosystem II and ATP synthase of the photosystem to regulate photosynthesis. The common proteins were still concentrated in the leaves to regulate photosynthesis.

According to the photosynthesis proteins, the KEGG pathway of the COS group had more proteins than that of the CK group. Under normal growth, COS could increase the expression of common proteins and promote photosynthesis. The salt stress stage had different regulation modes according to different parts, but COS could alleviate the adverse effects of salt stress to some extent. There were more unique proteins produced in rice stem, especially photosystem II and ATP synthase.

The above studies indicated that COS has a positive effect on rice’s resistance to salt stress and is environmentally friendly. With global climate change and the increasing scarcity of land resources, cultivating salt-tolerant rice is crucial. In practical rice breeding programs, we can use COS as a biological pesticide for rice or other crops to withstand salt stress, thereby enhancing the utilization of saline-alkali land and increasing rice yield.

## 3. Materials and Methods

### 3.1. Plant Materials and Rice Culture

Rice (Nipponbare) seeds (purchased from Xiantao Ruideweien Technology Co., Ltd.) (Xiantao, China) pretreatment: 3% sodium hypochlorite solution was used to disinfect seeds for 5 min. After that, the seeds were washed with ddH_2_O thoroughly. Then, the seeds were soaked in distilled water at 2 °C for 24 h, and they were transferred to the seedling plate and covered with wet seedling paper. The rice seeds were germinated under the temperature of 28 °C and humidity of 75%.

After rice seeds germinated, rice seedlings were put into Hoagland nutrient solution for hydroponic culture. A quantity of 150 mM NaCl was selected for the salt stress treatment [33,34,35]. Then, 150 mM NaCl was added at the three-leaf stage of rice growth [36,37], and the salt stress time was 5 days. During this period, the appropriate amount of NaCl was added to the nutrient solution to reach a concentration of 150 mmol/L. The mixed nutrient solution with added NaCl was replaced once daily. Finally, NaCl was removed, and the recovery time was 5 days. Throughout the seedling growth, the incubator was constant at 28 °C, 75% humidity, 12 h light and 12 h dark. Hoagland nutrient solution and salt solution were replaced every 2 days.

Treatment method of experimental group: 1.25 g/L COS (Golden Shell Pharmaceutical Co., Ltd. (Taizhou, China)) solution was sprayed when the rice germinated and grew to one leaf stage, and the spraying frequency was every day.

### 3.2. Determination of the Phenotypic Shape and Physiological Indicators

Samples were taken before salt stress, 5 d after salt stress and 5 d after recovery. Pictures were taken to observe the growth of the rice seedlings. Then, 30 rice seedlings were randomly selected to determine their root length, leaf length, plant height and leaf width. Ten rice plants were randomly selected in the seedling tray, and the dry weight and fresh weight of the roots, stems and leaves of rice seedlings were measured. Fresh rice seedling samples (0.2 g) were immersed in 95% ethanol, which was to extract chlorophyll for 12 h in complete darkness. Debris in the extract was separated by filtration through filter paper, and the debris was rinsed with ethanol several times. The chlorophyll content was then measured according to the methods described in Shen et al. [38].

Soluble sugar content was measured using the anthrone colorimetric method [39]. Leaves of rice were weighed and then extracted by 80% ethanol for 30 min with occasional agitation. The superior liquid was filtered and the volume was adjusted to 10 mL with 80% ethanol. A quantity of 1 mL of extract was incubated with 5 mL of anthrone reagent at 95 °C for 15 min, and then the reaction was terminated in an ice bath. Absorbance was measured at 620 nm.

The malondialdehyde (MDA) content was measured using the method described by Meng et al. [40]. Leaf tissue (0.2 g) was homogenized in 4 mL of 10% (*w*/*v*) trichloroacetic acid (TCA) with a mortar. After centrifugation at 5000 rpm for 10 min, 2 mL of supernatant with 2 mL of 0.6% thiobarbituric acid (TBA, 0.6% in 10% TCA) was mixed, heated for 15 min, and then quickly cooled and centrifuged at 5000 rpm for 10 min. The control contained 2 mL TCA instead of MDA extract. Absorbance was determined at 450, 532 and 600 nm.

Total amino acid content in rice seedlings was extracted using water and ninhydrin methods [41]. Rice roots, stems and leaves (0.2 g) were separately taken. Then, 10 mL of distilled water was added and boiled for 20 min. The mixture was then cooled down with cold water and filtered to separate the liquid, which was then boiled for another 10 min. The volume was adjusted to 25 mL, and the solution was mixed well to obtain the amino acid extraction solution. A quantity of 0.5 mL of acetic acid-NaCN buffer and water hydantoin (water as control) were added to the 0.5 mL of the extraction solution, then treated with boiling water for 12 min before cooling. After that, 5 mL of 95% ethanol was added and shaken well. The optical density value was measured at a wavelength of 570 nm.

SOD was measured using the NBT method [42]. Samples (0.1 g) were ground and homogenized with 2 mL of extracting solution (50 mmol/L PBS (pH = 7.8), 0.1M EDTA, 0.1% (*v*/*v*) Triton X-100, 2% (*w*/*v*) PVP) on ice. The homogenate was centrifuged at 3500× *g* for 15 min. Then, 0.1 mL of the supernatant liquor was added into the reaction system. The reaction was started by the addition of Met and riboflavin. The reaction tubes were placed beside a set of 4000 lx fluorescent tubes for 15 min. Then, the reaction system was measured with a spectrophotometer at 560 nm.

POD was measured using the method proposed by Yu et al. [43]. The preparation method for crude POD enzyme extract was the same as that for amino acid extract. Then, 1.5 mL of 0.05 M phosphate buffer and 130 mM methionine solution were added. The mixture was placed under a 4000 Lx fluorescent light for 30 min (the control group was kept in the dark to avoid light), with the reaction temperature ranging from 24–32 °C. The optical density value was measured at a wavelength of 560 nm.

CAT was measured using the method described by Li et al. [44]. The preparation of the CAT crude enzyme extract followed the same method as for the amino acid extract. The system (1.5 mL of 0.2 M phosphate buffer, 1 mL of distilled water and 0.2 mL of crude enzyme solution) was placed in a 25 °C water bath for 10 min, followed by the addition of 0.3 mL of 0.1 M H_2_O_2_ to initiate the reaction. The control group used inactivated enzyme. The absorbance was measured at a wavelength of 240 nm.

### 3.3. Protein Extraction, Proteolysis and Desalination

A quantity of 1.0 g of fresh roots, stems and leaves was placed in pre-cooled mortars, and 10% PVPP (polyvinyl pyrrolidone) was added. Then, liquid nitrogen was added to grind each part of the rice to powder. The samples were homogenized with the buffer of 10 mM Tris-HCl (pH 8.0) containing 8 M urea and 1 mM PMSF in the ice bath. The solution was transferred to a centrifuge tube, ultrasonically crushed for half an hour and centrifuged at 10,000 rpm for 20 min at 4 °C. The supernatant was then transferred to a new centrifuge tube and mixed with 20 mL of cold acetone (10% TCA and 10 mM DTT). The mixture was cooled to −20 °C, and protein was allowed to precipitate for 12 h before being centrifuged at 10,000 rpm for 20 min at 4 °C. After incubation at −20 °C for 4 h, the suspension was centrifuged at 10,000 rpm for 15 min at 4 °C. The supernatant was discarded, and the resulting pellet was washed twice with cold acetone (10 mM DTT).

After air-drying, 100 mM DTT was added to 1 mg/L protein solution, and the reaction mixture was incubated at 37 °C for 2 h. A quantity of 100 mM IAA was added to the solution for light avoidance reaction for 40 min at room temperature. Then, 10 mM Tris and 1 mg/mL trypsin (mass ratio 1:30) was added, and the protein was enzymatically digested for 12 h overnight in a 37 °C water bath.

The digested sample was passed through a C18 solid-phase extraction column. The effluent of the column was collected and put into a vacuum centrifugal concentrator for rotary drying. Then, 50 μL of 0.1% FA was added to redissolve the dry peptide and the prepared samples were subjected to mass spectrometry analysis.

### 3.4. LC-MS/MS Analysis

Chromatographic condition was as follows: mobile stage A: 0.1% FA (formic acid); mobile stage B: 0.1% FA (formic acid) + 80% ACN (acetonitrile); flow rate: 0.3 μL/min; column temperature: 45 °C; sample intake: 1 μg; precolumn: C18PepMap100 (5 μm, 100 Å); analytical column: AcclaimPepMapTMRSLC (75 μm × 15 cm, 3 μm). Flow-stage linear elution procedure was as follows: 0–10 min (4%B); 10–13 min (4–10%B); 13–90 min (10–35%B); 90–114 min (35–55%B); 114–115 min (55~95%B); 115–122 min (95%B); 122–123.5 min (95–4%B); 123.5–140 min (4%B).

Samples were ionized by nanoelectrospray (Nano ESI) and analyzed by mass spectrum of Q Exactive™ combined quadrupole Orbitrap mass spectrometer. The MS scanning mode was Full MS/dd-MS^2^. Primary MS conditions were as follows: Orbitrap resolution: 70,000; AGC: 3e^6^; maximum injection time: 100 ms; scan range: 350–1800 *m*/*z*. Secondary MS conditions were as follows: Orbitrap resolution: 17,500; AGC: 1e^5^; maximum injection time: 110 ms; number of parent ions: 15; collision energy: 27%.

RAW file data output by the mass spectrometry were searched using Proteome Discoverer (2.2.0.388) software. The total rice database (*Oryza sativa*, 1,303,332 sequences) obtained from Uniprot was used. Trypsin was a lyase; the maximum leakage site was 2. The mass error for precursor ions and fragment ions were set to 10 ppm and 0.02 Da, respectively. The results were analyzed by quantitative proteomics using the LFQ.

### 3.5. Label-Free Quantitative Proteomic Analysis

The LFQ results were screened with CV < 20% to obtain the stably expressed protein. Three comparison methods were selected to study the DEPs, namely: salt stress/normal growth (S/N), recovery/normal growth (R/N), recovery/salt stress (R/S). The relative change of 1.2 (CV < 20%) was up-regulated protein and the relative change of 0.8 (CV < 20%) was down-regulated protein.

### 3.6. GO Enrichment Analysis, Functional Annotation

By GO enrichment analysis (http://geneontology.org accessed on 9 April 2022), the resulting DEPs were annotated in three categories: biological process (BP), cellular component (CC) and molecular function (MF).

### 3.7. KEGG Pathway Analysis

Differential protein metabolism pathway analysis was performed using the online analysis platform KAAS (https://www.genome.jp/tools/kaas/ accessed on 20 June 2022).

## 4. Conclusions

After spraying COS, rice seedlings grew better, and the content of chlorophyll, soluble sugar and total amino acid, as well as root activity, in rice leaves were markedly promoted under salt stress. At the same time, the accumulation of the antioxidant enzymes SOD, POD and CAT was significantly increased with COS treatment, and the production of malondialdehyde was inhibited. According to quantitative proteomic analysis of the rice by GO annotation, more annotations were identified in the rice seedlings treated with COS after salt stress. The expression of differential proteins significantly changed, indicating that the expressed proteins are capable of resisting damage caused by salt stress. KEGG was used to interpret the two key pathways of glycolysis and photosynthesis. The key common proteins that were identified as chaning after the addition of COS include phosphate fructose kinase, glyceraldehyde-3-phosphate dehydrogenase, enolase and pyruvate kinase, as well as the oxygen-evolution-enhancing proteins PsbO, PsbP and PsbQ, along with the PetF and PetH proteins. These proteins play important roles in rice resistance to salt stress. The COS selected in this study are commercially available, and their application to crops shows promising prospects. Future research can explore various aspects such as COS concentration, duration, degree of polymerization, application timing, application methods and plant growth stages to optimize the conditions for using COS and reduce economic costs.

## Figures and Tables

**Figure 1 ijms-25-05953-f001:**
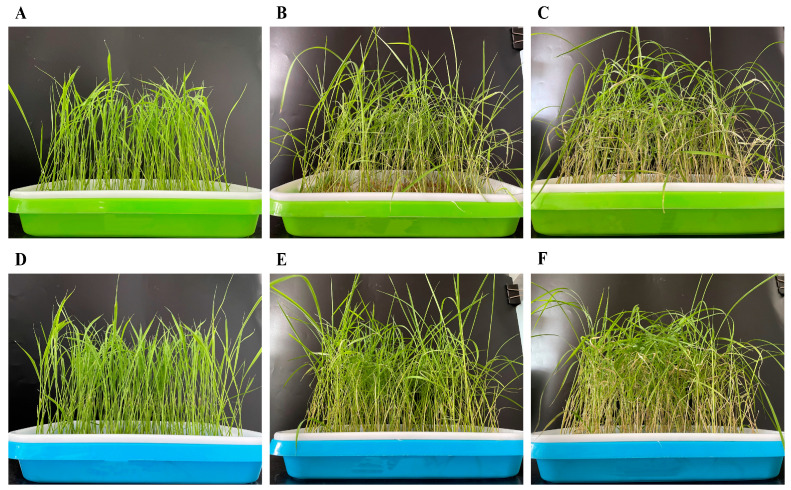
Growth of Nipponbare rice seedlings. (**A**–**C**) The growth of Nipponbare rice seedlings in the CK group. (**D**–**F**) The growth of Nipponbare rice seedlings in the COS group. (**A**,**D**) Normal growth. (**B**,**E**) 150 mM NaCl stress. (**C**,**F**) Recovery.

**Figure 2 ijms-25-05953-f002:**
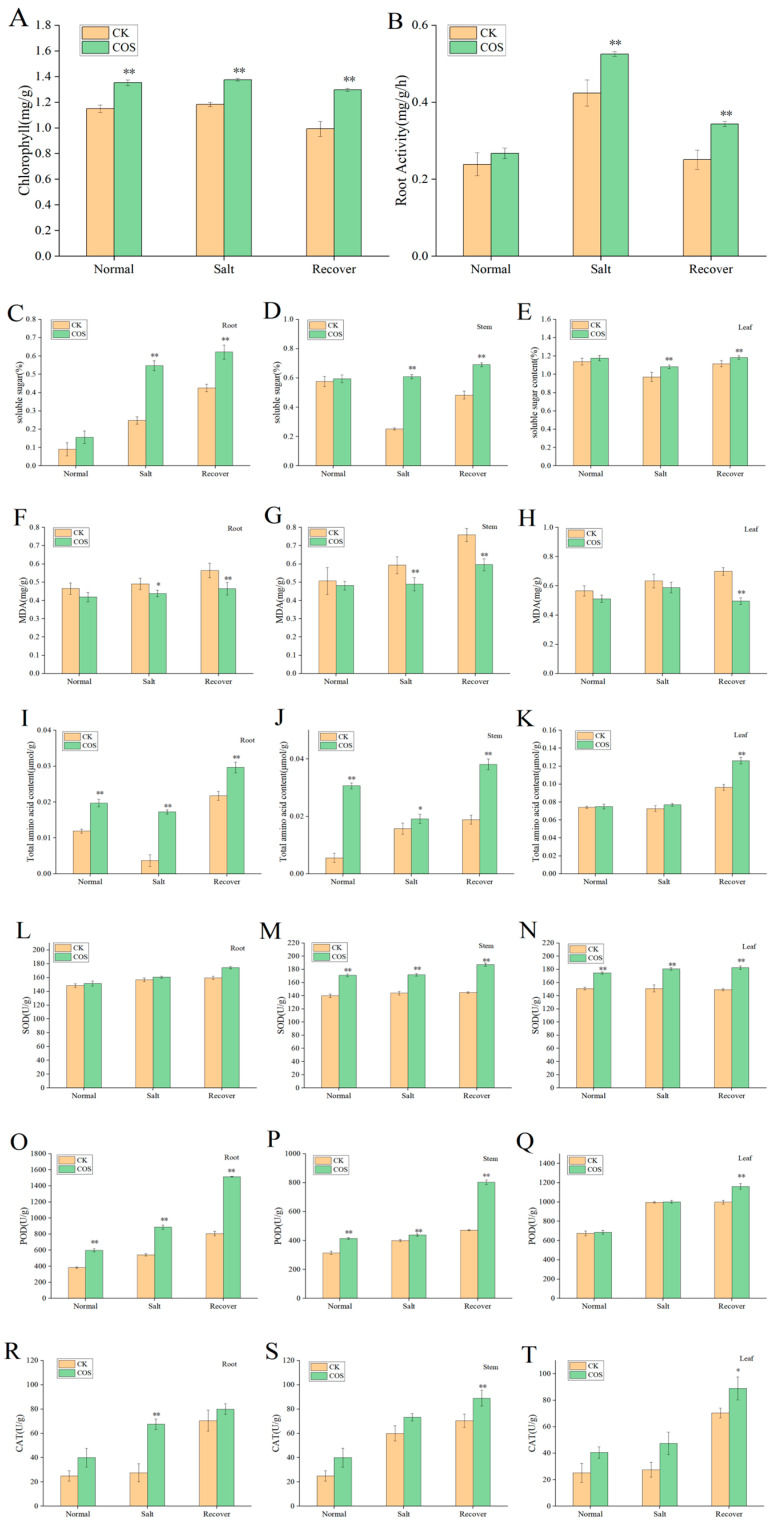
Biochemical indicators of the three stages of the Nipponbare rice seedlings. (**A**) Chlorophyll content, (**B**) root activity, (**C**–**E**) soluble sugar content, (**F**–**H**) malondialdehyde content, (**I**–**K**) total amino acid content, (**L**–**N**) SOD activity, (**O**–**Q**) POD activity, (**R**–**T**) CAT active. (**C**–**T**) A group of three biochemical indicators, including the three parts of the roots, stems and leaves, respectively. * *p* < 0.05; ** *p* < 0.01.

**Figure 3 ijms-25-05953-f003:**
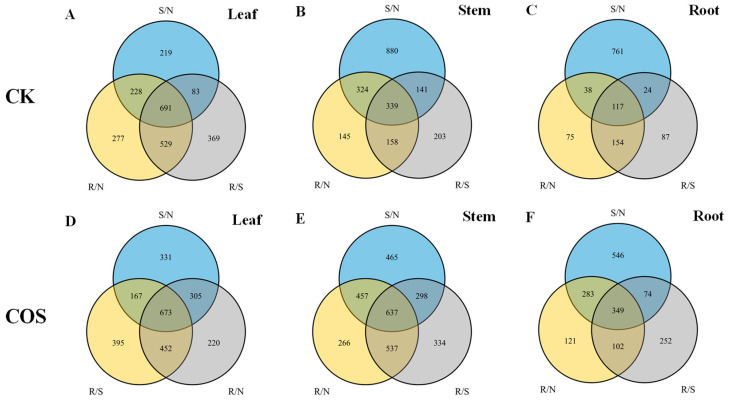
Venn diagram of DEPs between CK group and COS group in the rice seedlings of Nipponbare. (**A**–**C**) Venn diagrams of roots, stems and leaves of the CK group. (**D**–**F**) Venn diagrams of roots, stems and leaves of the COS group. S/N: salt stress/normal growth, R/N: recover/normal growth, R/S: recover/salt stress.

**Figure 4 ijms-25-05953-f004:**
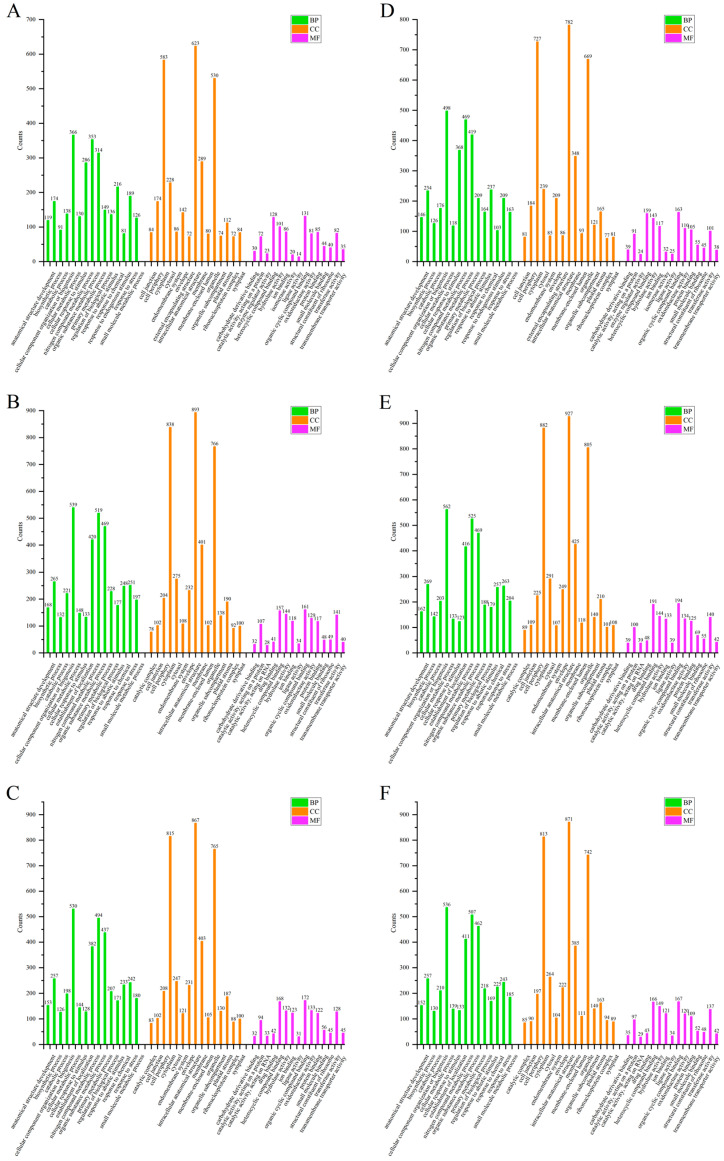
GO annotation of DEPs in the rice leaves of Nipponbare. (**A**–**C**) CK groups. (**A**) S/N salt stress/normal growth; (**B**) R/N recover/normal growth; (**C**) R/S recover/salt stress. (**D**–**F**) COS groups. (**D**) S/N salt stress/normal growth; (**E**) R/N recover/normal growth; (**F**) R/S recover/salt stress. BP: biological process, CC: cellular component, MF: molecular function.

**Figure 5 ijms-25-05953-f005:**
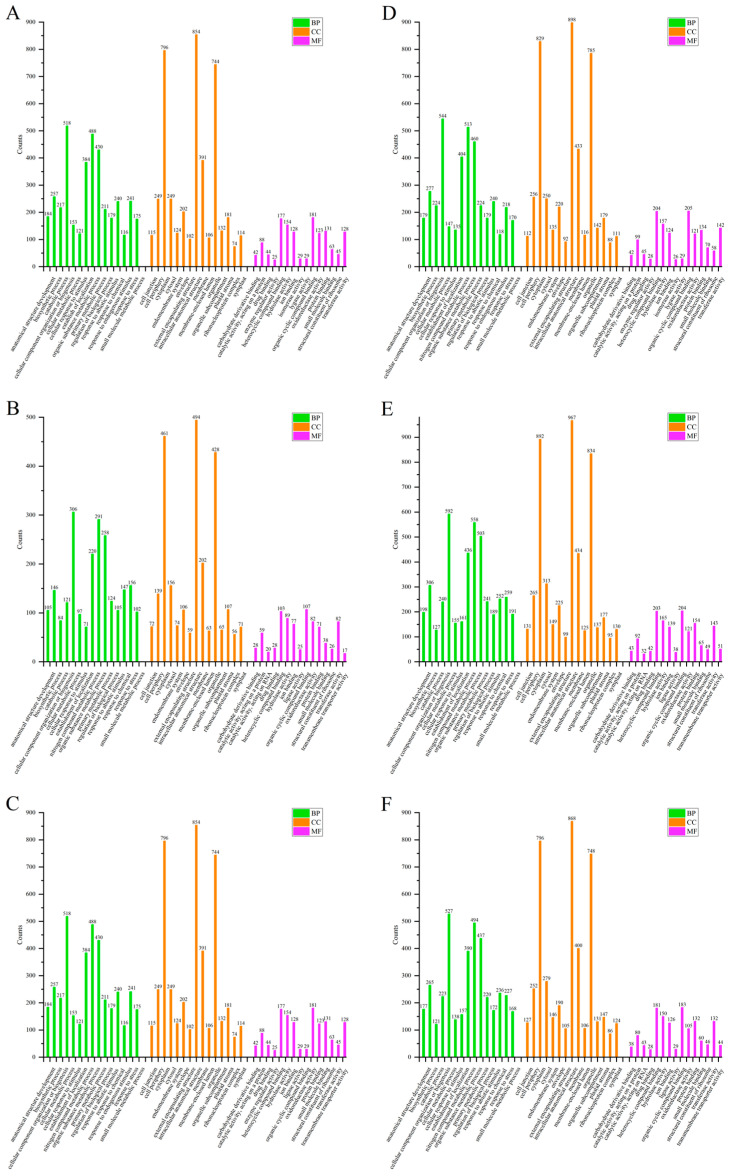
GO annotation of DEPs in the rice stems of Nipponbare. (**A**–**C**) CK groups. (**A**) S/N salt stress/normal growth; (**B**) R/N recover/normal growth; (**C**) R/S recover/salt stress. (**D**–**F**) COS groups. (**D**) S/N salt stress/normal growth; (**E**) R/N recover/normal growth; (**F**) R/S recover/salt stress. BP: biological process, CC: cellular component, MF: molecular function.

**Figure 6 ijms-25-05953-f006:**
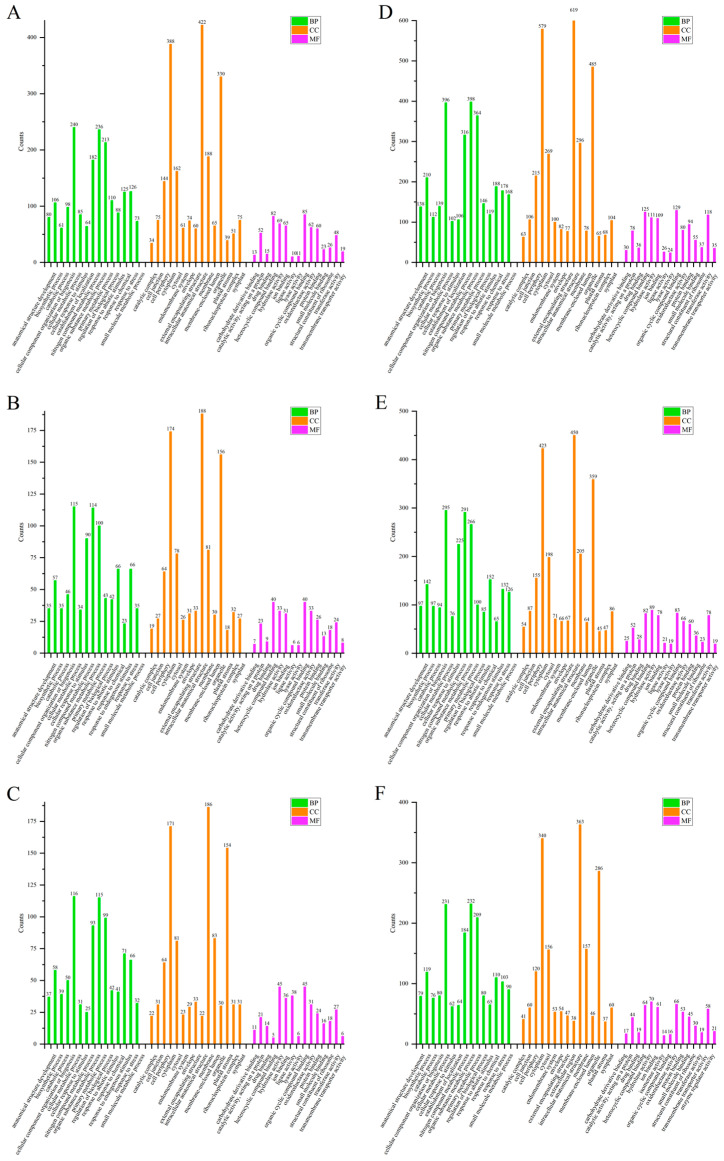
GO annotation of DEPs in the rice roots of Nipponbare. GO annotation of DEPs in the rice leaves of Nipponbare. (**A**–**C**) CK groups. (**A**) S/N salt stress/normal growth; (**B**) R/N recover/normal growth; (**C**) R/S recover/salt stress. (**D**–**F**) COS groups. (**D**) S/N salt stress/normal growth; (**E**) R/N recover/normal growth; (**F**) R/S recover/salt stress. BP: biological process, CC: cellular component, MF: molecular function.

**Figure 7 ijms-25-05953-f007:**
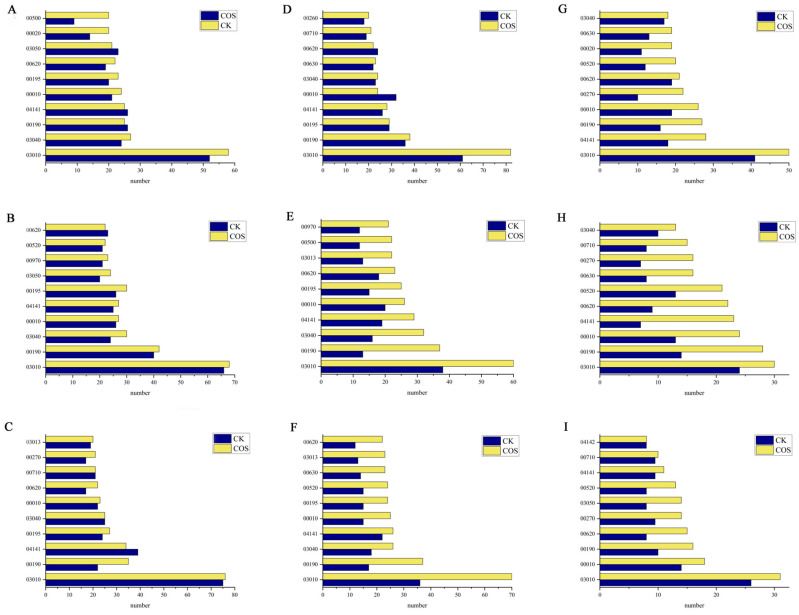
Top 10 key pathways for DEP enrichment in the rice seedlings of Nipponbare. (**A**–**I**) The enrichment of leaves ((**A**) S/N (**B**) R/N (**C**) R/S), stems ((**D**), S/N (**E**) R/N (**F**) R/S) and roots ((**G**) S/N (**H**) R/N (**I**) R/S) under salt stress/normal, recover/normal, and recover/salt stress stage. 00500: starch and sucrose metabolism, 00020: citrate cycle, 03050: proteasome, 00620: pyruvate metabolism, 00195: photosynthesis, 00010: glycolysis/gluconeogenesis, 04141: protein processing in endoplasmic reticulum, 00190: oxidative phosphorylation, 03040: splice body, 03010: ribosome, 00520: amino sugar and nucleotide sugar metabolism, 00970: aminoacyl-tRNA biosynthesis, 03013: nuclear and cytoplasmic transport, 00270: cysteine and methionine metabolism, 00710: carbon sequestration in photosynthetic organisms, 00260: metabolism of glycine, serine and threonine, 00630: glyoxylic acid and dicarboxylic acid metabolism, 04142: lysosome.

**Table 1 ijms-25-05953-t001:** Phenotypic parameters of rice seedlings of Nipponbare.

		Normal	Stress			Recovery	
		CK	COS	CK	COS	CK	COS
Plant height(cm)	root length	6.71 ± 0.95	6.82 ± 0.80	6.90 ± 1.03	8.47 ± 1.34 **	8.23 ± 1.22	9.61 ± 1.53 **
	leaf length	11.06 ± 2.09	12.17 ± 2.38	12.75 ± 2.51	15.48 ± 1.76 **	18.70 ± 2.11	20.65 ± 1.41 **
	overall length	19.09 ± 2.03	19.54 ± 2.61	21.34 ± 2.35	24.53 ± 1.48 **	29.42 ± 2.40	32.10 ± 2.42 **
	leaf width	0.29 ± 0.04	0.32 ± 0.03 *	0.30 ± 0.03	0.32 ± 0.02	0.24 ± 0.05	0.29 ± 0.03 **
Fresh weight(g)	roots	0.237 ± 0.015	0.290 ± 0.017 *	0.290 ± 0.020	0.310 ± 0.030	0.270 ± 0.010	0.290 ± 0.010
	stems	0.453 ± 0.021	0.503 ± 0.012 *	0.487 ± 0.085	0.573 ± 0.035	0.520 ± 0.010	0.593 ± 0.021 **
	leaves	0.210 ± 0.017	0.227 ± 0.015	0.353 ± 0.035	0.367 ± 0.038	0.377 ± 0.00	0.390 ± 0.010
Dry weight(g)	roots	0.021 ± 0.002	0.024 ± 0.004	0.023 ± 0.003	0.025 ± 0.002	0.020 ± 0.003	0.021 ± 0.001
	stems	0.145 ± 0.002	0.153 ± 0.003 *	0.144 ± 0.005	0.158 ± 0.003 **	0.102 ± 0.004	0.134 ± 0.007 **
	leaves	0.038 ± 0.001	0.041 ± 0.001 *	0.069 ± 0.008	0.069 ± 0.006	0.079 ± 0.008	0.080 ± 0.002

* *p* < 0.05; ** *p* < 0.01.

**Table 2 ijms-25-05953-t002:** Number of proteins identified by mass spectrometry in the rice seedlings of Nipponbare.

Position	Normal		Stress		Recover	
	CK	COS	CK	COS	CK	COS
Leaf	2326	2618	2711	2795	2712	2794
	2371	2605	2675	2785	2687	2752
	2394	2593	2681	2784	2680	2781
Stem	2012	2948	2122	2913	2185	2645
	1976	2863	1990	2900	2099	2673
	2004	2824	1948	2852	1963	2612
Root	1997	2380	2549	2271	2100	2079
	1987	2316	2515	2189	2045	2065
	1990	2353	2505	2087	2023	2054

**Table 3 ijms-25-05953-t003:** Detailed information of the number of stable proteins, differentially expressed proteins, up-regulated proteins and down-regulated proteins in the rice seedlings of Nipponbare.

Position	Condition	Stage	SPs	DEPs	UP	DOWN
Leaf	CK	S/N	2285	1221	1067	154
		R/N	2206	1725	`1156	569
		R/S	2378	1672	1039	633
	COS	S/N	2427	1476	1006	470
		R/N	2236	1650	1060	590
		R/S	1460	1687	1027	660
Stem	CK	S/N	2599	1684	928	756
		R/N	1437	966	589	377
		R/S	1555	841	576	265
	COS	S/N	3227	1857	1082	775
		R/N	2853	1897	922	975
		R/S	2976	1806	829	977
Root	CK	S/N	2110	940	198	742
		R/N	495	384	186	198
		R/S	527	382	181	201
	COS	S/N	1595	1252	435	817
		R/N	1084	855	258	597
		R/S	1316	778	335	443

SPs are stably present proteins, DEPs are differentially expressed proteins, UP are up-regulated proteins, DOWN are down-regulated proteins, S/N: salt stress/normal growth, R/N: recovery/normal growth, R/S: recovery/salt stress.

**Table 4 ijms-25-05953-t004:** Common proteins produced by the CK group and COS group in the glycolysis pathway of Nipponbare rice seedlings.

Position	Name	KO		Abundance			Description
				N	S	R	
Leaf	5.3.1.9	K0180	CK	105.4	92.9	101.8	glucose-6-beta-phosphate isomerase
			COS	89.3	115.3	95.4	
	4.1.2.13	K01623	CK	68	95	136.9	fructose-bisphosphate aldolase
			COS	68.4	95	136.6	
	1.2.1.12	K00134	CK	50.3	77.4	172.3	glyceraldehyde 3-phosphate dehydrogenase (phosphorylating)
			COS	70.7	109.6	119.7	
	1.2.1.9	K00131	CK	80.1	99.5	120.4	glyceraldehyde-3-phosphate dehydrogenase (NADP+)
			COS	78.7	110	111.3	
	4.2.1.11	K01689	CK	78.5	113.4	108.0	enolase
			COS	66.7	124.3	108.9	
	2.7.1.40	K00873	CK	46.7	63.8	166.1	pyruvate kinase
			COS	67.2	109.1	123.8	
	2.7.9.1	K01006	CK	78.6	87.9	133.5	pyruvate, orthophosphate dikinase
			COS	60.8	104.5	134.6	
	1.2.4.1	K00163	CK	77.3	92.3	130.3	pyruvate dehydrogenase E1 component
			COS	70.6	106.2	123.2	
	2.3.1.12	K00627	CK	80.6	103.9	115.6	pyruvate dehydrogenase E2 component
			COS	82.6	106.8	110.7	
	1.8.1.4	K00382	CK	87.9	89.8	122.4	dihydrolipoyl dehydrogenase
			COS	75.7	111.4	112.9	
	1.2.1.3	K00128	CK	63.4	82.8	153.9	aldehyde dehydrogenase
			COS	74.1	103.8	122.3	
	1.1.1.1	K13951	CK	34.9	70.9	194.2	alcohol dehydrogenase
			COS	44.9	76.6	178.5	
Stem	4.1.2.13	K01623	CK	101.2	88.2	110.6	fructose-bisphosphate aldolase
			COS	91.3	89.5	119.3	
	5.4.2.12	K15633	CK	140	71.9	88	2,3-bisphosphoglycerate-independent phosphoglycerate mutase
			COS	142.6	98.9	58.5	
	2.7.1.40	K00873	CK	158.3	65.7	76.1	pyruvate kinase
			COS	122.6	73.9	103.4	
	2.7.9.1	K01006	CK	76.1	116.6	107.3	orthophosphate dikinase
			COS	76.8	112.7	110.5	
	1.2.4.1	K00163	CK	90.8	92.0	117.3	pyruvate dehydrogenase E1 component
			COS	99.6	96.2	104.2	
	1.1.1.1	K13951	CK	78.4	114.0	120.9	alcohol dehydrogenase
			COS	78.8	107.3	113.9	
Root	2.7.1.90	K00895	CK	102.7	130.9	66.4	diphosphate-dependent phosphofructokinase
			COS	143.4	80.4	76.3	
	4.1.2.13	K01623	CK	123.8	62.3	127.9	fructose-bisphosphate aldolase
			COS	128.9	89.2	82.1	
	2.7.2.3	K00927	CK	148.8	82.1	69	phosphoglycerate kinase
			COS	170.2	61.5	68.3	
	5.4.2.12	K15633	CK	160	104.8	35.2	2,3-bisphosphoglycerate-independent phosphoglycerate mutase
			COS	153.3	72.8	73.9	
	2.7.1.40	K00873	CK	102.8	116.5	80.7	pyruvate kinase
			COS	152.8	82.7	64.6	
	1.8.1.4	K00382	CK	83.6	48.4	168	dihydrolipoyl dehydrogenase
			COS	112.7	107.8	79.5	
	1.2.1.3	K00128	CK	70.7	111.9	102.8	aldehyde dehydrogenase
			COS	87.2	84.0	128.9	
	1.1.1.1	K13951	CK	59.2	124.2	116.7	alcohol dehydrogenase
			COS	68.2	117.9	113.9	

**Table 5 ijms-25-05953-t005:** Differential proteins produced by the COS group in the glycolysis pathway in the rice seedlings of Nipponbare.

Position	S/N			R/N		
	Name	KO	Accession	Name	KO	Accession
Leaf	1.1.1.2	K00002	A0A0D9WH90	1.1.1.2	K00002	B9F274
	5.4.2.2	K01835	A0A0E0D5L5	1.1.1.2	K00002	A0A0D9WH90
	2.7.1.1	K00844	A0A0E0A1S8	4.1.1.49	K01610	A0A0E0CY82
	3.1.3.11	K03841	Q9SDL8	6.2.1.1	K01895	Q7F8W1
	5.3.1.1	K01803	A0A0E0B5S4	5.4.2.12	K15633	B8AZB8
	5.3.1.1	K01803	P48494	5.4.2.12	K15633	A0A0E0FUX9
	5.4.2.12	K15633	A0A0E0FUX9			
	4.1.1.49	K01610	A0A0E0CY82			
Stem				2.7.1.1	K00844	A0A0D3G422
				2.7.1.1	K00844	A0A0E0DZ42
				5.1.3.3	K01785	Q33AZ5
				5.1.3.3	K01785	A0A0P0WB27
				5.1.3.15	K01792	A2YZX3
				2.7.1.90	K00895	I1P3K4
				2.7.1.90	K00895	A0A0E0PVR9
				4.1.1.49	K01610	A0A0E0CY82
				6.2.1.1	K01895	Q7F8W1
				1.8.1.4	K00382	B9FMK1
				1.8.1.4	K00382	A0A0D9V0H3
Root	5.4.2.2	K01835	A0A0E0D5L3	2.7.1.1	K00844	A0A0D9YES9
	2.7.1.1	K00844	Q5W676	5.3.1.9	K01810	A0A0E0M263
	2.7.1.1	K00844	A0A0D3ETH5	5.1.3.3	K01785	A0A0E0HWM4
	3.1.3.11	K03841	A0A0D3G850	2.7.1.11	K24182	A0A0D9Y4F5
	2.7.1.11	K24182	J3N2I3	4.2.1.11	K01689	I1QSV0
	2.7.1.11	K24182	A0A0D9Y4F5	4.2.1.11	K01689	B8AK24
	2.7.1.11	K24182	A0A0P0WBF4	4.2.1.11	K01689	A0A0E0B2A0
	1.2.1.12	K00134	A0A0D9YMY6	1.2.4.1	K00163	A0A0E0QTJ9
	1.2.1.12	K00134	B8AF09	4.1.1.1	K01568	A0A0D3FH89
	1.2.1.9	K00131	Q8S4Y9	2.3.1.12	K00627	A0A0D3GPZ8
	4.1.1.49	K01610	A3AG67	1.8.1.4	K00382	B9FMK1
	4.1.1.1	K01568	A0A0D3FH89			

**Table 6 ijms-25-05953-t006:** Common proteins produced by the CK group and COS group in the photosynthetic pathway of Nipponbare rice seedlings.

Position	Name	KO		Abundance			Description
				N	S	R	
Leaf	PsbQ	K02713	CK	115.1	47.6	58.8	photosystem II PsbL protein
			COS	193.5	74.2	110.7	
	PsbO	K02716	CK	118.7	124.6	113.6	photosystem II oxygen-evolving enhancer protein 1
			COS	153.6	89.0	57.5	
	PsbP	K02717	CK	56.6	75.4	148.1	photosystem II oxygen-evolving enhancer protein 2
			COS	75.7	77.1	147.3	
	Psb28	K08903	CK	48.2	70.1	181.7	photosystem II 13kDa protein
			COS	48.2	105.2	146.6	
	PsaB	K02690	CK	64.8	82.7	152.6	photosystem I P700 chlorophyll a apoprotein A2
			COS	79.7	88.1	132.2	
	PsaC	K02691	CK	95.5	120.2	84.3	photosystem I subunit VII
			COS	83.5	112	104.5	
	PsaE	K02693	CK	36.5	70.5		photosystem I subunit IV
			COS	80.8	109.6	109.6	
	PsaG	K08905	CK	85	89	125.9	photosystem I subunit V
			COS	80.1	113	106.9	
	PsaK	K02698	CK	48.5	72.5	178.9	photosystem I subunit X
			COS	52.2	84.8	163.1	
	PsaL	K02699	CK	61	76.7	162.3	photosystem I subunit XI
			COS	72.3	91.3	136.5	
	PetF	K02639	CK	142.8	122.4	34.9	ferredoxin
			COS	148.4	102.3	49.2	
	PetH	K02641	CK	115.4	113.7	70.9	ferredoxin--NADP+ reductase
			COS	61.8	95	143.3	
	PetH	K02641	CK	20.6	58.1	221.3	ferredoxin--NADP+ reductase
			COS	19.5	111.8	168.6	
Stem	PsbO	K02716	CK	82.6	110.7	106.7	photosystem II oxygen-evolving enhancer protein 1
			COS	84.2	97.8	117.9	
	PsbP	K02717	CK	92.8	104.9	102.4	photosystem II oxygen-evolving enhancer protein 2
			COS	95.3	107.1	97.6	
	PsbS	K03542	CK	123.4	114.9	61.7	photosystem II 22kDa protein
			COS	75.8	103.7	120.5	
	PsaG	K08905	CK	74.1	145.5	80.5	photosystem I subunit V
			COS	82.5	119.8	97.6	
	PsaN	K02701	CK	128.8	77.9	93.3	photosystem I subunit PsaN
			COS	105	65.4	129.6	
	PetF	K02639	CK	102	137.6	60.5	ferredoxin
			COS	71	135.3	93.6	
	PetH	K02641	CK	66	82.2	151.8	ferredoxin—NADP+ reductase
			COS	141.3	80.5	78.2	

**Table 7 ijms-25-05953-t007:** Differential proteins produced by the COS group in the photosynthetic pathway in the rice seedlings of Nipponbare.

Position	S/N			R/N		
	Name	KO	Accession	Name	KO	Accession
Leaf	alpha	K02111	Q8S7T5	PsbE	K02707	E9KIX2
		K02113	Q6Z8K7	PsaJ	K02697	A0A1W5HPA3
	a	K02108	A0A0E0EPX3	PetB	K02635	A0A0H3V867
	b	K02109	A0A0E0CYU2	gamma	K02115	I1QB20
	PsbA	K02703	C5MRL2	a	K02108	A0A0E0EPX3
	PsaH	K02695	A2Y7D9	b	K02109	A0A0E0CYU2
	PetB	K02635	A0A0H3V867			
	PetD	K02637	A0A172GFK2			
Stem	PsbB	K02704	A0A0H3V867	beta	K02112	Q2QW50
	PsbE	K02707	E9KIX2	beta	K02112	A0A172GDI6
	PsbH	K02709	E9KJ46	a	K02108	A0A0E0EPX3
	Psb28	K08903	A0A0D3EYG3	b	K02109	A0PJ32
	PsaL	K02699	Q2QSR5	b	K02109	A0A0E0CYU2
	PetE	K02638	A0A0D3GBR0	alpha	K02111	Q8S7T5
				PsbD	K02706	E9KJ14
				PsbB	K02704	A0A1W5HNT3
				PsbB	K02704	A0A0H3V867
				PsbE	K02707	E9KIX2
				PsbF	K02708	E9KIQ3
				PsbH	K02709	E9KJ46
				PsbR	K03541	A0A0E0E7C7
				PsaB	K02690	E9KIP0
				PsaD	K02692	A0A0E0AYY8
				PsaE	K02693	A0A0E0LJV8
				PetB	K02635	A0A0H3V867
				PetE	K02638	A0A0D3GBR0

## Data Availability

The data presented in this study are available in the article.

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
