# Peer review of "Physiological and Proteome Analysis of the Effects of Chitosan Oligosaccharides on Salt Tolerance of Rice Seedlings"

_ijms, 2024, doi:10.3390/ijms25115953_

Round 1

Reviewer 1 Report

Comments and Suggestions for Authors

The researchers investigated the effects of chitosan oligosaccharides (COS) on salt resistance of rice seedlings through physiological and proteome analysis. They found that the application of COS can enhance the resistance of rice seedlings to salt stress. The manuscript can be accepted after addressing my below comments.

Title

·       The title is succinct but can convey the main topics of the study.

·       “…salt resistance…” or “…salt tolerance…”?. At my best understanding, the term “resistance” is commonly used for biotic stresses like pest or disease resistance.

Abstract

·       The abstract covers justification, objective, methods, and key findings, but it lacks a recommendation statement. I suggest authors add recommendations from their study, for instance, implications for molecular breeding programs.

Keywords

·       The keywords are informative and cover the overall work of the study.

Introduction:

·       In the first paragraph (line 36-49), authors have provided sufficient information regarding the importance of rice, salt stress, and its adverse effects on rice production worldwide.

·       In the second paragraph (line 50-61), authors have provided sufficient information regarding the importance of COS and proteome analysis. However, it still lacks two points such as (i) the source of COS materials and (ii) the recent application of proteome analysis for similar studies in rice or other related crop species. Please add them.

·       In the third paragraph (line 62-73), authors mentioned the rice cultivar Nipponbare as plant materials for the study; however, they did not explain the justification of that cultivar. Please add brief information regarding rice var. Nipponbare.

·       This part lacks research gap, hypothesis, and expected outcomes. Please add them in the last paragraph.

Materials and Methods:

·       Explain the source of rice materials var. Nipponbare.

·       Why did authors choose 150 mM NaCl instead of other concentrations? Did the authors conduct the preliminary study to determine that concentration? Please explain and provide the references if it is possible.

·       Is the three-leaf stage correct for rice screening? Please explain and provide the references if it is possible.

·       Please expand the procedure for imposing the salt stress in the experiment.

Discussions:

·       The discussion is well-written but can be further improved by providing the implications of all parameters observed in practical breeding programs.

·       In the section of “Growth and morphological characteristics of rice seedlings” and “Biochemical indicators of rice seedlings”, authors only provide the comparison review between their findings and previous reports. It would be better if authors could discuss comprehensively the mechanisms of salt stress regarding those parameters.

Conclusion:

·       Conclusion can address all research objectives mentioned in the manuscript.

References

·       All citations are listed in the references, update, and relevant to the research topic.

Comments on the Quality of English Language

Minor editing of English language required

Author Response

For research article

Response to Reviewer 1 Comments

1. Summary

Thank you very much for taking the time to review this manuscript. We have made the corresponding modifications based on your comments. Specific answers to each question are provided below and please find the detailed responses below. The parts that have been corrected are highlighted in red font, and some questions have also been explained accordingly. You can also see the corresponding corrections in the newly submitted manuscript.

2. Questions for General Evaluation

Reviewer’s Evaluation

Response and Revisions

Does the introduction provide sufficient background and include all relevant references?

Can be improved

Are all the cited references relevant to the research?

Yes

Is the research design appropriate?

Can be improved

Are the methods adequately described?

Can be improved

Are the results clearly presented?

Yes

Are the conclusions supported by the results?

Yes

3. Point-by-point response to Comments and Suggestions for Authors

Comments 1:

The title is succinct but can convey the main topics of the study.

“…salt resistance…” or “…salt tolerance…”?. At my best understanding, the term “resistance” is commonly used for biotic stresses like pest or disease resistance.

Response 1:

Thank you for pointing this out. We agree with this comment. Therefore, We have revised the title. We changed the title from “Physiological and proteome analysis of the effects of chitosan oligosaccharides on salt resistance of rice seedlings” to “ Physiological and proteome analysis of the effects of chitosan oligosaccharides on salt tolerance of rice seedlings”. You can review it in the front page of the revised manuscript.

Comments 2:

The abstract covers justification, objective, methods, and key findings, but it lacks a recommendation statement. I suggest authors add recommendations from their study, for instance, implications for molecular breeding programs.

Response 2:

Agree. We made a recommended statements based on the study. The modifications are made as follows. Furthermore, using COS as biopesticides and biofertilizers can effectively increase the utilization of saline-affected farmland, thereby contributing to the alleviating of the global food crisis. You can find it in lines 32-34 of the newly revised manuscript.

Comments 3:

In the second paragraph (line 50-61), authors have provided sufficient information regarding the importance of COS and proteome analysis. However, it still lacks two points such as (i) the source of COS materials and (ii) the recent application of proteome analysis for similar studies in rice or other related crop species. Please add them.

Response 3:

According to (i) to this question, we made the following supplement. Chitosan is a linear polysaccharide derived from the N-deacetylation of chitin, containing varying degrees of N-acetylation. It consists primarily of β -(1,4)-linked 2-acetamido-D-glucopyranose and 2-amino-D-glucopyranose units, with the latter typically comprising more than 80% of the structure [9]. Chitosan is naturally synthesized by fungi in their cell walls. COS is a type of chitosan with a low degree of polymerization (≤ 20) and an average molecular weight below 3900 Da, usually falling within the range of 0.2-3.0 kDa [10]. You can find it in lines 52-58 of the newly revised manuscript.

According to (ii), we have also supplemented this question accordingly. For example, Sengupta utilized proteomics to uncover over 400 protein spots that significantly altered under salt stress across different salt-tolerant rice germplasm. Among these differentially expressed proteins, a highly upregulated protein in wild salinity-tolerant rice, named SSP8908, was identified as a cellulose synthase-like protein under 400 mM NaCl stress. SSP8908 likely serves as a crucial factor in regulating cellulose synthesis in the cell walls of wild rice, thereby potentially enhancing its salt tolerance [16]. Liu conducted a proteomic analysis to compare ubiquitin-related proteins involved in salt sensitivity in the root systems of the TNG67 rice variety and the NaN3-induced pure mutants SM75 (salt-tolerant) and SA0604 (salt-sensitive). The results indicated that under salt stress, ubiquitination of cellulose synthase α increased in TNG67 and SM75 but decreased in SA0604. Moreover, the salt tolerance of these lines correlated with the abundance of ubiquitinated cellulose synthase α under salt stress, further highlighting the significant role of cellulose in the salt tolerance of rice [17]. Consequently, proteomic studies are helpful in revealing changes in protein abundance and the effects of different protein modifications in plants under various stresses. You can find it in lines 68-82 of the newly revised manuscript.

Comments 4:

In the third paragraph (line 62-73), authors mentioned the rice cultivar Nipponbare as plant materials for the study; however, they did not explain the justification of that cultivar. Please add brief information regarding rice var. Nipponbare.

This part lacks research gap, hypothesis, and expected outcomes. Please add them in the last paragraph.

Response 4:

We have made the following changes based on your question.

Oryza Sativa L spp. japonica, var Nipponbare serves as a crucial model organism for the study of rice genetic material. Through sequential cloning techniques, Nipponbare is the only rice variety among monocots whose genome has been sequenced with high quality. The completion of its genome sequencing has facilitated further study of rice gene functions, contributing to food security [18]. Therefore, this experiment was performed using rice (Nipponbare) as an experimental material, and a comparison was made between the experimental group (COS) sprayed with COS and the CK group without COS. You can find it in lines 83-89 of the newly revised manuscript.

Comments 5

Explain the source of rice materials var. Nipponbare.

Response 5:

The following supplements are made from your question.

Rice (Nipponbare) seeds (purchased from Xiantao Ruideweien Technology Co., Ltd.) (Hubei, China) pretreatment: 3% sodium hypochlorite solution was used to disinfect seeds for 5 min. You can find it in lines 470-472 of the newly revised manuscript.

Comments 6

Why did authors choose 150 mM NaCl instead of other concentrations? Did the authors conduct the preliminary study to determine that concentration? Please explain and provide the references if it is possible.

Response 6:

Because at the beginning of the experiment, we conducted a extensive literature search on the use of salt concentration. Here are the supplements we did. 150mM NaCl was selected for the salt stress treatment [33-35]. You can find it in line 477 of the newly revised manuscript.

33. Yan, S.; Tang, Z.; Su, W.; Sun, W. Proteomic analysis of salt stress‐responsive proteins in rice root. Proteomics 2005, 5, 235-244.

34. Cheng, Y.; Qi, Y.; Zhu, Q.; Chen, X.; Wang, N.; Zhao, X.; Chen, H.; Cui, X.; Xu, L.; Zhang, W. New changes in the plasma‐membrane‐associated proteome of rice roots under salt stress. Proteomics 2009, 9, 3100-3114.

35. Xu, J.; Lan, H.; Fang, H.; Huang, X.; Zhang, H.; Huang, J. Quantitative proteomic analysis of the rice (Oryza sativa L.) salt response. PLoS One 2015, 10, e0120978.

Therefore, after comprehensive consideration, we chose this concentration for the experiment.

Comments 7

Is the three-leaf stage correct for rice screening? Please explain and provide the references if it is possible.

Response 7:

Yes. I have made a literature inquiry on this. It is described in two articles. Here are the supplements we did. Then 150 mM NaCl was added at the three-leaf stage of rice growth [36-37], and the salt stress time was 5 days. You can find it in lines 478-479 of the newly revised manuscript.

36. Almeida, D.M.; Almadanim, M.C.; Lourenço, T.; Abreu, I.A.; Saibo, N.J.; Oliveira, M.M. Screening for abiotic stress tolerance in rice: salt, cold, and drought. Environmental Responses in Plants: Methods and Protocols 2016, 155-182.

37. Chang, T.S.; Liu, C.W.; Lin, Y.L.; Li, C.Y.; Wang, A.Z.; Chien, M.W.; Wang, C.S.; Lai, C.C. Mapping and comparative proteomic analysis of the starch biosynthetic pathway in rice by 2D PAGE/MS. Plant molecular biology 2017, 95, 333-343.

Comments 8

Please expand the procedure for imposing the salt stress in the experiment.

Response 8:

The salt stress stage is specified according to the above problem.

During this period, the appropriate amount of NaCl was added to the nutrient solution to reach a concentration of 150 mmol/L. The mixed nutrient solution with added NaCl was replaced once daily. You can find it in lines 478-481 of the newly revised manuscript.

Comments 9

The discussion is well-written but can be further improved by providing the implications of all parameters observed in practical breeding programs.

Response 9:

According to your suggestions, we have made the following supplement. The above studies indicated that COS has a positive effect on rice's resistance to salt stress and is environmentally friendly. With global climate change and the increasing scarcity of land resources, cultivating salt-tolerant rice is crucial. In practical rice breeding programs, we can mass-produce COS as a biological pesticide for rice or other crops to withstand salt stress, thereby enhancing the utilization of saline-alkali land and increasing rice yield. You can find it in lines 458-462 of the newly revised manuscript.

Comments 10

In the section of “Growth and morphological characteristics of rice seedlings” and “Biochemical indicators of rice seedlings”, authors only provide the comparison review between their findings and previous reports. It would be better if authors could discuss comprehensively the mechanisms of salt stress regarding those parameters.

Response 10:

In the section of “Growth and morphological characteristics of rice seedlings”,although we described the roots, stems and leaves separately, we also summarized the results of these three parts in the article. For instance the lines 112-113 in the article: Therefore, it could be seen that spraying COS could improve the harm of salt stress to rice on some extent. This is a comprehensive description of the rice morphology. The lines 112-113 in the article: The results align with Liu and Mostek's research, showing that salt stress could affect the dry and fresh weights of roots, stems, and leaves, and that after salt stress, both the dry and fresh weights of rice seedlings increased [19, 20]. This sentence summarizes the indicators of fresh weight and dry weight of the three parts.

In the section of “Biochemical indicators of rice seedlings”. The following modifications were made by following your recommendations. SOD can catalyze superoxide radicals to hydrogen peroxide (H2O2) and that POD can scavenge H2O2. CAT can protect cells from H2O2 damage by catalyzing the decomposition of H2O2 into O2 and H2O. In general, the enhanced activities of SOD, POD, and CAT were rice’s defense responses to long-term salt stress, and the addition of COS enhanced these defense responses. You can find it in lines 170-174 of the newly revised manuscript.

4. Response to Comments on the Quality of English Language

Point 1: Minor editing of English language required

Response 1:

According to your opinion, we have made appropriate modifications to the English language . In the content added, we refer to some literature for careful consideration.

5. Additional clarifications

Response:

No.

Reviewer 2 Report

Comments and Suggestions for Authors

The manuscript entitled “Physiological and proteome analysis of the effects of chitosan oligosaccharides on salt resistance of rice seedlings” aimed to investigate the effects of chitosan oligosaccharides on the growth and morphological characteristics, as well as biochemical indicators of rice seedlings under salt stress conditions. Additionally, a quantitative proteomics analysis was conducted to assess the differential protein expression of rice seedlings treated with chitosan oligosaccharides under both control and stress conditions. This study involved extensive experimental work, and the methodology employed was deemed adequate. The conclusions drawn were well-supported by the obtained results. The manuscript is well-written and offers a significant contribution to the field. Based on these considerations, I recommend accepting this manuscript after the authors address minor revisions.

-Lines 46-47: this statement needs to be supported by a reference.

-The quality of Figure needs to be improved.

-In plant material section more information about the used rice cultivar (Nipponbare) is required.

-The choice of 150 mM NaCl as salt stress for rice should be justified.

Author Response

For research article

Response to Reviewer 2 Comments

1. Summary

Thank you very much for taking the time to review this manuscript. We have made the corresponding modifications based on your comments. Specific answers to each question are provided below and please find the detailed responses below. The parts that have been corrected are highlighted in red font, and some questions have also been explained accordingly. You can also see the corresponding corrections in the newly submitted manuscript.

2. Questions for General Evaluation

Reviewer’s Evaluation

Response and Revisions

Does the introduction provide sufficient background and include all relevant references?

Can be improved

Are all the cited references relevant to the research?

Yes

Is the research design appropriate?

Yes

Are the methods adequately described?

Yes

Are the results clearly presented?

Yes

Are the conclusions supported by the results?

Yes

3. Point-by-point response to Comments and Suggestions for Authors

Comments 1:

Lines 46-47: this statement needs to be supported by a reference.

Response 1: Thank you for pointing this out. Based on your suggestion, we have conducted literature supplementation. The most sensitive period to salt stress is during the seedling stage, and the subsequent vegetative growth period will improve its tolerance to salt stress [7-8]. You can find it in line 50 of the newly revised manuscript.

Comments 2:

The quality of Figure needs to be improved.

Response 2:

Following your suggestion, we have made changes to Figures 2,4,5,6, and 7.

Comments 3:

In plant material section more information about the used rice cultivar (Nipponbare) is required.

Response 3:

The following supplements are made from your question.

Rice (Nipponbare) seeds (purchased from Xiantao Ruideweien Technology Co., Ltd.) (Hubei, China) pretreatment: 3% sodium hypochlorite solution was used to disinfect seeds for 5 min. You can find it in lines 470-471 of the newly revised manuscript.

Comments 4:

The choice of 150 mM NaCl as salt stress for rice should be justified.

Response 4:

Because at the beginning of the experiment, we conducted a extensive literature search on the use of salt concentration. Here are the supplements we did. 150mM NaCl was selected for the salt stress treatment [33-35]. You can find it in line 477 of the newly revised manuscript.

33. Yan, S.; Tang, Z.; Su, W.; Sun, W. Proteomic analysis of salt stress‐responsive proteins in rice root. Proteomics 2005, 5, 235-244.

34. Cheng, Y.; Qi, Y.; Zhu, Q.; Chen, X.; Wang, N.; Zhao, X.; Chen, H.; Cui, X.; Xu, L.; Zhang, W. New changes in the plasma‐membrane‐associated proteome of rice roots under salt stress. Proteomics 2009, 9, 3100-3114.

35. Xu, J.; Lan, H.; Fang, H.; Huang, X.; Zhang, H.; Huang, J. Quantitative proteomic analysis of the rice (Oryza sativa L.) salt response. PLoS One 2015, 10, e0120978.

Therefore, after comprehensive consideration, we chose this concentration for the experiment.

4. Response to Comments on the Quality of English Language

Point 1: English language fine. No issues detected

Response 1:

Thank you for your comment. In the content added, we refer to some literature for careful consideration.

5. Additional clarifications

Response:

No.

Reviewer 3 Report

Comments and Suggestions for Authors

In the manuscript "Physiological and proteome analysis of the effects of chitosan oligosaccharides on salt resistance of rice seedlings’’ Qian and collaborators studied the effect of chitosan oligosaccharides on salinity tolerance in rice by physiological and proteomic analyses, concluding that COS might alleviate salinity stress in rice seedlings.

Overall, the manuscript addresses the initial objectives of the work, and it is written in an appropriate way. However, there are many sentences that need correction and/or clarification. These are some examples but there are many more:

Line 31-32: should be rewritten for clarity

Line 62: the authors did not perform whole genome sequencing analysis… should be rewritten for correction

Line 127 and throughout the text: what do the authors mean by ‘root activity’?... should be rewritten for clarity

Line 129: ‘COS could resist the damage of salt stress’ … should be rewritten for clarity

Line 179: what do the authors mean by ‘geographically different’…. should be rewritten for clarity

Line 272: what do the authors mean by ‘related to the life activities’ …. should be rewritten for clarity

Line 452 – 456: there is no need to include this in the manuscript

Line 464: should be rewritten for clarity

Line 468 – 474: the authors should describe the methods, at least briefly, indicating possible modifications done, and not just refer to the references

Line 535: should be rewritten for clarity

Line 540-542: should be rewritten for clarity

The Results and Discussion section present clearly the results, it is well written, and comparisons are made with the findings obtained in other related works. The authors present the conclusions of their work which are supported by the data shown, however future work is missing.

Author Response

For research article

Response to Reviewer 3 Comments

1. Summary

Thank you very much for taking the time to review this manuscript. We have made the corresponding modifications based on your comments. Specific answers to each question are provided below and please find the detailed responses below. The parts that have been corrected are highlighted in red font, and some questions have also been explained accordingly. You can also see the corresponding corrections in the newly submitted manuscript.

2. Questions for General Evaluation

Reviewer’s Evaluation

Response and Revisions

Does the introduction provide sufficient background and include all relevant references?

Must be improved

Are all the cited references relevant to the research?

Can be improved

Is the research design appropriate?

Can be improved

Are the methods adequately described?

Can be improved

Are the results clearly presented?

Can be improved

Are the conclusions supported by the results?

Yes

3. Point-by-point response to Comments and Suggestions for Authors

Comments 1:

Line 31-32: should be rewritten for clarity.

Response 1:

Thank you for pointing this out. We agree with this comment. Therefore, we made the changes. The addition of COS led to an increase in the abundance of proteins, a response of rice seedlings to salt stress. COS helped rice seedlings resist salt stress. You can find it in lines 31-32 of the newly revised manuscript.

Comments 2:

Line 62: the authors did not perform whole genome sequencing analysis… should be rewritten for correction

Response 2:

We have made the following changes by following your suggestion. Therefore, this experiment was performed using rice (Nipponbare) as an experimental material, and a comparison was made between the experimental group (COS) sprayed with COS and the CK group without COS. You can find it in lines 87-89 of the newly revised manuscript.

Comments 3:

Line 127 and throughout the text: what do the authors mean by ‘root activity’?... should be rewritten for clarity

Response 3:

Root activity is the ability of roots to take up water and nutrients, and their ability to respond to external environmental conditions. Specific instructions of root vitality are available in this literature. ‘Methods to estimate changes in soil water for phenotyping root activity in the field’. The measurement of root activity in this study is conducted to gain a more comprehensive understanding of the changes in various physiological and biochemical indicators of rice seedlings after the addition of COS, thereby arriving at more accurate conclusions.

Comments 4:

Line 129: ‘COS could resist the damage of salt stress’ … should be rewritten for clarity

Response 4:

Make correction according to your recommendations. It indicated that the addition of COS could enhance the root activity of rice seedlings under salt stress to maintain their normal growth (Figure 2B). You can find it in lines 153-155 of the newly revised manuscript.

Comments 5:

Line 179: what do the authors mean by ‘geographically different’…. should be rewritten for clarity

Response 5:

Here is what we have changed. This showed that under salt stress, the protein expression in rice seedlings shows positional differences and salt stress also caused significant changes in rice protein expression in the whole plant. You can find it in lines 208-210 of the newly revised manuscript.

Comments 6:

Line 272: what do the authors mean by ‘related to the life activities’ …. should be rewritten for clarity

Response 6:

 Here is what we have changed. It showed that after rice seedlings were subjected to salt stress, in order to resist the damage from salt stress, its vital activities have changed, which in turn led to alterations in annotations. You can find it in lines 300-302 of the newly revised manuscript.

Comments 7:

Line 452 – 456: there is no need to include this in the manuscript

Response 7:

We have deleted this paragraph according to your suggestion.

Comments 8:

Line 464: should be rewritten for clarity

Response 8:

 Here is what we have changed. The chlorophyll content was then measured according to the methods described in Shen et al [38]. You can find it in lines 496-497 of the newly revised manuscript.

Comments 9:

Line 468 – 474: the authors should describe the methods, at least briefly, indicating possible modifications done, and not just refer to the references

Response 9:

The following additions were made following your recommendations.

The chlorophyll content was then measured according to the methods described in Shen et al [38].

Soluble sugar content was measured using the anthrone colorimetric method [39]. Leaves of rice were weighed and then extracted by 80% ethanol for 30 min with occasional agitation. The superior liquid was filtered and the volume was adjusted to 10 mL with 80% ethanol. 1 mL of extract was incubated with 5 mL of anthrone reagent at 95°C for 15 min, and then the reaction was terminated in an ice bath. The absorbance was measured at 620 nm.

The malondialdehyde (MDA) content was measured using the method described by Meng et al [40]. Leaf tissue (0.2 g) was homogenized in 4 mL of 10% (w/v) trichloroacetic acid (TCA) with a mortar. After centrifugation at 5000 rpm for 10 min, 2 mL of supernatant with 2 mL of 0.6% thiobarbituric acid (TBA, 0.6% in 10% TCA) was mixed, heated for 15 min, and then quickly cooled and centrifuged at 5000 rpm for 10 min. The control contained 2 mL TCA instead of MDA extract. Absorbance was determined at 450, 532, and 600 nm.

Total amino acid content in rice seedlings was extracted using water and ninhydrin methods [41]. Rice roots, stems, and leaves (0.2g) were separately taken. Then 10ml of distilled water was added and boiled for 20 minutes. The mixture was then cooled down with cold water, filtered to separate the liquid, which was continued to be boiled for another 10 minutes. The volume was adjusted to 25ml, and the solution was mixed well to obtain the amino acid extraction solution. 0.5ml of acetic acid-NaCN buffer and water hydantoin (water as control) were added to the 0.5ml of the extraction solution, then treated with boiling water for 12 minutes before cooling. After that, 5ml of 95% ethanol was added and shaken well. The optical density value was measured at a wavelength of 570nm.

SOD was measured using the NBT method [42]. Samples (0.1 g) were ground and homogenized with 2mL of extracting solution (50mmol/L PBS (pH= 7.8), 0.1M EDTA, 0.1% (v/v) Triton X-100, 2% (w/v) PVP) on ice. The homogenate was centrifuged at 3,500 g for 15min. Then 0.1mL of the supernatant liquor was added into the reaction system. The reaction was started by the addition of Met and riboflavin. The reaction tubes were placed beside a set of 4000 lx fluorescent tubes for 15min. Then the reaction system was measured with a spectrophotometer at 560nm.

POD was measured using the method proposed by Yu et al [43]. The preparation method for crude POD enzyme extract was the same as that for amino acid extract. Then, 1.5mL of 0.05M phosphate buffer and 130mM methionine solution were added. The mixture was placed under a 4000Lx fluorescent light for 30 minutes (the control group was kept in the dark to avoid light), with the reaction temperature ranging from 24-32°C. The optical density value was measured at a wavelength of 560nm.

CAT was measured using the method described by Li et al [44]. The preparation of the CAT crude enzyme extract followed the same method as for the amino acid extract. The system (1.5mL of 0.2M phosphate buffer, 1mL of distilled water, and 0.2mL of crude enzyme solution) was placed in a 25°C water bath for 10 minutes, followed by the addition of 0.3mL of 0.1M H2O2 to initiate the reaction. The control group used inactivated enzyme. The absorbance was measured at a wavelength of 240nm.

You can find it in lines 496-538 of the newly revised manuscript.

Comments 10:

Line 535: should be rewritten for clarity

Response 10:

We made the following changes following your advice. According to quantitative proteomic analysis of the rice by GO annotation, more annotations were identified in the rice seedlings treated with COS after salt stress. And the expression of differential proteins significantly changed, indicating that the expressed proteins are capable of resisting damage caused by salt stress.You can find it in lines 597-600 of the newly revised manuscript.

Comments 11:

Line 540-542: should be rewritten for clarity

Response 11:

We made the following changes following your advice. According to quantitative proteomic analysis of the rice by GO annotation, more annotations were identified in the rice seedlings treated with COS after salt stress. And the expression of differential proteins significantly changed, indicating that the expressed proteins are capable of resisting damage caused by salt stress. KEGG was used to interpret the two key pathways of glycolysis and photosynthesis. And the key common proteins that were identified to change after the addition of COS include phosphate fructose kinase, glyceraldehyde-3-phosphate dehydrogenase, enolase, pyruvate kinase, as well as the oxygen evolution enhancing proteins PsbO, PsbP, and PsbQ, along with the PetF and PetH proteins. These proteins play important roles in rice resistance to salt stress. The COS selected in this study are commercially available, and their application to crops shows promising prospects. Future research can explore various aspects such as COS concentration, duration, degree of polymerization, application timing, application methods, and plant growth stages to optimize the conditions for using COS and reduce economic costs.

You can find it in lines 597-610 of the newly revised manuscript.

Comments 12: The Results and Discussion section present clearly the results, it is well written, and comparisons are made with the findings obtained in other related works. The authors present the conclusions of their work which are supported by the data shown, however future work is missing.

Response 12:

According to your suggestions, we have made the following supplement. The above studies indicated that COS has a positive effect on rice's resistance to salt stress and is environmentally friendly. With global climate change and the increasing scarcity of land resources, cultivating salt-tolerant rice is crucial. In practical rice breeding programs, we can mass-produce COS as a biological pesticide for rice or other crops to withstand salt stress, thereby enhancing the utilization of saline-alkali land and increasing rice yield. You can find it in lines 458-462 the newly revised manuscript.

4. Response to Comments on62 o the Quality of English Language

Point 1: I am not qualified to assess the quality of English in this paper.

Response 1:

Thank you for your advice. In the content added, we refer to some literature for careful consideration.

5. Additional clarifications

Response:

No.

Reviewer 4 Report

Comments and Suggestions for Authors

Thank you for contacting me to review this manuscript by Xiangyu Qia, et al.  In this work the authors evaluated the effect of Chitosan oligosaccharide (COS) on the rice seedlings growth and development. Authors studied the salt tolerance mechanism on rice seedlings exposed to COS by measuring different parameters such as protein content, chlorophyll content, biochemical indexes, etc during three stages normal growth, salt stress and recovery. The found that after treatment with COS, the chlorophyll content increased, the root activity during the recovery stage was higher than in CK group. Other factor measured such as soluble sugar in root, stem and leaf increased in plants exposed to COS. Authors concluded that COS help rice seeding in the resistance to salt stress.

Overall, the manuscript has several flaws. The introduction is short and general, the authors need to add more information about COS, factor triggered by salt stress, etc. The manuscript writing style need to be improved for a better understanding and some results does not support the conclusion the author stated.

There are issues enlisted below:

1) page 1, line 31: “All these results demonstrated that COS was demonstrated to resist the effects of salt stress on rice seedlings.” COS is in fact resisting the salt stress or is helping rice seedling to resist. Please make the sentence clear.

2) page 2, line 50: “COS has attracted wide attention because of its low molecular, short chain structure,” please defined COS acronym. Please completed the word low molecular weight.

3) page 2 : please specific CK control, GO, KEEG acronyms

4) page 2, line 71: “…was then performed using differential proteins, and protein functions and pathways were…” I think differential proteins can be changed to “differential protein expression or differential protein content.”

5) page 3, paragraph 1: “The CK group exhibited more severe lodging and yellowing compared to the COS group” according to the figure 1, the phenotype in CK group is not more severe than in COS, it is a slight difference.

6) page 3, paragraph 1: “In the recovery stage, as shown in Figure 1C and F, rice tips and stems in both groups were more yellow than during the salt stress stage” This statement is not supported by the Fig 1. I could not see a significant difference in salt stress vs recovery stage. Was there any quantitative method to support the statement or it was only a qualitative analysis?

7) page 3, line 95: it is mentioned that COS spraying increases fresh weight. Is there a control using just water for spraying the control group?

8) page 6, line 161: fix the typo error “Lable-free Quantitative” to “Label-free Quantitative”

9) page 6, line 163: specify S/N and R/N

10) page 8, line 228 : fix “Go annotation”  to “GO annotation”

11) page 18, line 442: please include the seller reference and catalog number

12) provide higher resolution figures for Fig2, Fig4, Fig 5, Fig6, and Fig7,

13) Conclusion is very short, please provide more information, details, etc

Comments on the Quality of English Language

English language shoud be improved and coherence is required for better understanding 

Author Response

For research article

Response to Reviewer 4 Comments

1. Summary

Thank you very much for taking the time to review this manuscript. We have made the corresponding modifications based on your comments. Specific answers to each question are provided below and please find the detailed responses below. The parts that have been corrected are highlighted in red font, and some questions have also been explained accordingly. You can also see the corresponding corrections in the newly submitted manuscript.

2. Questions for General Evaluation

Reviewer’s Evaluation

Response and Revisions

Does the introduction provide sufficient background and include all relevant references?

Must be improved

Are all the cited references relevant to the research?

Must be improved

Is the research design appropriate?

Must be improved

Are the methods adequately described?

Must be improved

Are the results clearly presented?

Must be improved

Are the conclusions supported by the results?

Must be improved

3. Point-by-point response to Comments and Suggestions for Authors

Comments 1:

page 1, line 31: “All these results demonstrated that COS was demonstrated to resist the effects of salt stress on rice seedlings.” COS is in fact resisting the salt stress or is helping rice seedling to resist. Please make the sentence clear.

Response 1:

Thank you for pointing this out. We agree with this comment. Therefore, we made the changes. The addition of COS led to an increase in the abundance of proteins, a response to salt stress. COS helped rice seedlings resist salt stress. You can find it in lines 31-32 of the newly revised manuscript.

Comments 2:

page 2, line 50: “COS has attracted wide attention because of its low molecular, short chain structure,” please defined COS acronym. Please completed the word low molecular weight.

Response 2: Agree. We made this statement and supplement the literature. The modifications are made as follows: Chitosan is a linear polysaccharide derived from the N-deacetylation of chitin, containing varying degrees of N-acetylation. It consists primarily of β -(1,4)-linked 2-acetamido-D-glucopyranose and 2-amino-D-glucopyranose units, with the latter typically comprising more than 80% of the structure [9]. Chitosan is naturally synthesized by fungi in their cell walls. COS is a type of chitosan with a low degree of polymerization (≤ 20) and an average molecular weight below 3900 Da, usually falling within the range of 0.2-3.0 kDa [10]. You can find it in lines 52-58 of the newly revised manuscript.

Comments 3:

page 2 : please specific CK control, GO, KEEG acronyms

Response 3:

According to this question, we made the following supplement. “ CK control” on line 65 was corrected as “blank group” on line 91. “GO” on line 72 was corrected as “Gene Ontology” on line 97. “KEEG” on line 73 was corrected as “Kyoto Encyclopedia of Genes and Genomes ” on lines 97.

Comments 4:

page 2, line 71: “…was then performed using differential proteins, and protein functions and pathways were…” I think differential proteins can be changed to “differential protein expression or differential protein content.”

Response 4:

We have made the following changes based on your question. Enrichment analysis was then performed using differential protein expression, and protein functions and pathways were analyzed using Gene Ontology and Kyoto Encyclopedia of Genes and Genomes.

You can find that in lines 95-97 of the newly revised manuscript.

Comments 5

page 3, paragraph 1: “The CK group exhibited more severe lodging and yellowing compared to the COS group” according to the figure 1, the phenotype in CK group is not more severe than in COS, it is a slight difference.

Response 5:

We have made the following changes based on your question. In comparison to the COS group, the CK group showed more pronounced lodging and yellowing. You can find it in lines 106-107 of the newly revised manuscript.

Comments 6

page 3, paragraph 1: “In the recovery stage, as shown in Figure 1C and F, rice tips and stems in both groups were more yellow than during the salt stress stage” This statement is not supported by the Fig 1. I could not see a significant difference in salt stress vs recovery stage. Was there any quantitative method to support the statement or it was only a qualitative analysis?

Response 6:

We have carefully considered your opinion. From this Figure 1, it may not be possible to see a significant difference between the CK group and the COS group due to the issue with the photos, but it is evident that in panels C and F, compared to B and E, the stems and leaves are yellower and the lodging is more pronounced. It is only a qualitative analysis.

Comments 7

page 3, line 95: it is mentioned that COS spraying increases fresh weight. Is there a control using just water for spraying the control group?

Response 7:

Yes. In this experiment, we only sprayed water on the control group to facilitate comparison with the COS group.

Comments 8

page 6, line 161: fix the typo error “Lable-free Quantitative” to “Label-free Quantitative”

Response 8:

Thank you for your correction. We have corrected it according to your suggestion. “Lable-free Quantitative” was corrected as “Label-free Quantitative”. You can find it in line 190 of the newly revised manuscript.

Comments 9

page 6, line 163: specify S/N and R/N

Response 9:

According to your suggestions, we have made the following supplement.

Of these, in S/N (salt stress/normal growth) and R/S (recovery /salt stress). You can find it in line 213 of the newly revised manuscript.

Comments 10

page 8, line 228 : fix “Go annotation” to “GO annotation”

Response 10:

We have made the following change. “Go annotation” was corrected as “GO annotation”

You can find that in line 257 of the newly revised manuscript.

Comments 11

page 18, line 442: please include the seller reference and catalog number

Response 11:

Thanks very much for your advice. Combining them with the opinions of the other reviewers, we describe the deletion of experimental reagents. But we supplemented the source of the seeds. Rice (Nipponbare) seeds (purchased from Xiantao Ruideweien Technology Co., Ltd.) (Hubei, China) pretreatment: 3% sodium hypochlorite solution was used to disinfect seeds for 5 min. You can find it in lines 470-471 of the newly revised manuscript.

Comments 12

provide higher resolution figures for Fig2, Fig4, Fig 5, Fig6, and Fig7

Response 12:

Following your suggestion, we have made changes to Figures 2,4,5,6, and 7.

Comments 13

Conclusion is very short, please provide more information, details, etc

Response 13:

We have made the following change. According to quantitative proteomic analysis of the rice by GO annotation, more annotations were identified in the rice seedlings treated with COS after salt stress. And the expression of differential proteins significantly changed, indicating that the expressed proteins are capable of resisting damage caused by salt stress. KEGG was used to interpret the two key pathways of glycolysis and photosynthesis. And the key common proteins that were identified to change after the addition of COS include phosphate fructose kinase, glyceraldehyde-3-phosphate dehydrogenase, enolase, pyruvate kinase, as well as the oxygen evolution enhancing proteins PsbO, PsbP, and PsbQ, along with the PetF and PetH proteins. These proteins play important roles in rice resistance to salt stress. The COS selected in this study are commercially available, and their application to crops shows promising prospects. Future research can explore various aspects such as COS concentration, duration, degree of polymerization, application timing, application methods, and plant growth stages to optimize the conditions for using COS and reduce economic costs. You can find that in lines 597-610 of the newly revised manuscript.

4. Response to Comments on the Quality of English Language

Point 1: English language shoud be improved and coherence is required for better understanding

Response 1:

According to your opinion, we have made appropriate modifications to the English language . In the content added, we refer to some literature for careful consideration.

5. Additional clarifications

Response:

No.

Round 2

Reviewer 4 Report

Comments and Suggestions for Authors

I do not have further suggestions or concerns